# Gab1 mediates PDGF signaling and is essential to oligodendrocyte differentiation and CNS myelination

Liang Zhou[1,2,3,4†], Chong-Yu Shao[1,2,3†], Ya-Jun Xie[1,2,3], Na Wang[5], Si-Min Xu[1,2,3], Ben-Yan Luo[1,2,3], Zhi-Ying Wu[6], Yue Hai Ke[7], Mengsheng Qiu[8], Ying Shen[1,2,3]*

[1]Department of Physiology of First Affiliated Hospital, Zhejiang University School of Medicine, Hangzhou, China; [2]Department of Neurology of First Affiliated Hospital, Zhejiang University School of Medicine, Hangzhou, China; [3]NHC and CAMS Key Laboratory of Medical Neurobiology, Zhejiang University School of Medicine, Hangzhou, China; [4]Key Laboratory of Brain Science, Guizhou Institution of Higher Education, Zunyi Medical University, Zunyi, China; [5]School of Medicine, Zhejiang University City College, Hangzhou, China; [6]Department of Neurology and Research Center of Neurology in Second Affiliated Hospital, Key Laboratory of Medical Neurobiology of Zhejiang Province, Zhejiang University School of Medicine, Hangzhou, China; [7]Department of Pathology and Pathophysiology, Zhejiang University School of Medicine, Hangzhou, China; [8]Institute of Life Sciences, Zhejiang Key Laboratory of Organ Development and Regeneration, College of Life and Environmental Sciences, Hangzhou Normal University, Hangzhou, China

*For correspondence:
yshen@zju.edu.cn

†These authors contributed equally to this work

Competing interests: The authors declare that no competing interests exist.

**Abstract** Oligodendrocytes (OLs) myelinate axons and provide electrical insulation and trophic support for neurons in the central nervous system (CNS). Platelet-derived growth factor (PDGF) is critical for steady-state number and differentiation of oligodendrocyte precursor cells (OPCs), but its downstream targets are unclear. Here, we show for the first time that Gab1, an adaptor protein of receptor tyrosine kinase, is specifically expressed in OL lineage cells and is an essential effector of PDGF signaling in OPCs in mice. Gab1 is downregulated by PDGF stimulation and upregulated during OPC differentiation. Conditional deletions of *Gab1* in OLs cause CNS hypomyelination by affecting OPC differentiation. Moreover, Gab1 binds to downstream GSK3β and regulated its activity, and thereby affects the nuclear accumulation of β-catenin and the expression of a number of transcription factors critical to myelination. Our work uncovers a novel downstream target of PDGF signaling, which is essential to OPC differentiation and CNS myelination.

## Introduction

In the central nervous system (CNS), oligodendrocytes (OLs) myelinate axons and provide electrical insulation and trophic support for neurons (*Simons and Nave, 2015*). The precursors of OLs, oligodendrocyte precursor cells (OPCs), are generated from the germinal regions of neural tube (*Rowitch, 2004*), and then proliferate and migrate throughout the CNS before differentiating into OLs. The proliferation, migration and differentiation of OPCs are coordinated in a predictable manner by numerous extrinsic and intrinsic factors (*Miller, 2002*; *Emery, 2010*), including axonally expressed ligands (*Wang et al., 1998*; *Charles et al., 2000*; *Mi et al., 2005*), nuclear transcription factors (*Fu et al., 2002*; *Arnett et al., 2004*; *Battiste et al., 2007*; *He et al., 2007*; *Emery et al., 2009*), and mitogens, for example, platelet-derived growth factor (PDGF) (*Pringle et al., 1992*), fibroblast growth factor (FGF) (*Furusho et al., 2017*), netrins and semaphorins (*Spassky et al., 2002*;

*Tsai et al., 2006*), and chemokine CXCL1 (*Filipovic and Zecevic, 2008*). Among these molecules, PDGF is a major in vivo mitogen for OPC development (*Pringle et al., 1992*; *Fruttiger et al., 1999*). PDGF provided by neurons and astrocytes determines the steady-state number of OPCs in the developing CNS (*van Heyningen et al., 2001*) and negatively regulates OPC differentiation. In cultures, the withdrawal of PDGF from the medium rapidly stops the proliferation and initiates the differentiation of OPCs (*Barres et al., 1993*). Correspondingly, the inactivation of PDGFα receptor (PDGFRα), which is majorly expressed in OPCs (*Pringle et al., 1992*), results in a reduced number of OPCs and precocious OPC differentiation (*Zhu et al., 2014*), whereas the activation of PDGFRα facilitates OPC division and migration (*Frost et al., 2009*; *Tripathi et al., 2017*). Although these studies demonstrate that PDGF serves as a gate controller of OPC development, it is surprising that the downstream targets of PDGF/PDGFRα signaling participating in OPC proliferation and differentiation are poorly understood.

The growth factor receptor bound 2 (Grb2)-associated binders, Gab1 and Gab2, are scaffolding proteins that act downstream of cell surface receptors, and interact with a variety of cytoplasmic signaling proteins, such as Grb2, SH2-containing protein tyrosine phosphatase 2 (Shp2), and phosphatidylinositol 3-kinase (PI3K) (*Liu and Rohrschneider, 2002*). It is known that Gab1 functions in lung diseases, such as allergic asthma and idiopathic pulmonary fibrosis, by interacting with cytoplasmic signaling proteins (*Wang et al., 2016*; *Zhang et al., 2016*; *Guo et al., 2017*). In the CNS, Gab proteins interact with growth factors, including epidermal growth factor (EGF), FGF, and PDGF, and modulate the mitotic process of neural progenitor cells (*Korhonen et al., 1999*; *Cai et al., 2002*; *Mao and Lee, 2005*). Interestingly, *Gab1* deletion in Schwann cells interrupts neuregulin-1 (NRG-1)-induced peripheral nerve myelination (*Shin et al., 2014*). However, the functions of Gab proteins in OL development and CNS myelination are not understood.

In the present study, we sought to investigate the functions of Gab proteins in mediating OPC differentiation and CNS myelination, given the interaction between growth factors and Gab proteins in neural progenitor cells and the importance of PDGF signaling in OL development. Our study provides compelling evidence that Gab1 is an important downstream effector of PDGF signaling during OPC differentiation and regulates CNS myelination by modulating the activity of GSK3β and β-catenin.

## Results

### Distinct effects of triiodothyronine and PDGF on Gab1 expression in OPCs

To investigate the roles of Gab proteins in OL development, we first assessed their expressions in oligodendrocyte linage cells and other types of neural cells. Using purified cultures, we uncovered a number of interesting findings: i) Gab1 and Gab2 were not uniformly expressed in neural cells. Gab1 was highly expressed in astrocytes and oligodendrocyte linage cells, whereas Gab2 was highly expressed in neurons, astrocytes and microglia (*Figure 1A*); ii) Gab1 was absent from cortical neurons (*Figure 1A*); and iii) Gab1 expression was remarkably elevated in mature OLs compared with OPCs (*Figure 1A*), accompanying by the increased expression of myelin-specific proteins, myelin basic protein (MBP) and myelin oligodendrocyte glycoprotein (MOG) (*Figure 1A and B*). The western blotting was corroborated by immunocytochemical staining, showing intense Gab1 signals in cell bodies and elaborated processes of mature OLs (*Figure 1C*).

It has been shown that the differentiation of cultured OPCs is promoted by triiodothyronine, but suppressed by PDGF (*Barres et al., 1993*). Consistently, we found that external PDGF-AA (10 ng/ml) treatment arrested OPC differentiation, as indicated by a much reduced increase in MBP expression (*Figure 1D*). Interestingly, either 1- or 3 day treatment with PDGF-AA significantly decreased Gab1 expression in OPCs (*Figure 1D*). To better evaluate the opposite effects of PDGF and triiodothyronine on Gab1, we administered PDGF-AA and triiodothyronine (40 ng/ml) simultaneously in OPCs. Our results demonstrated that PDGF-AA was sufficient to reverse the Gab1 expression augmented by triiodothyronine (*Figure 1D*). To confirm in vitro results, we assessed Gab1 expression in *Pdgfra* conditional knockout (*Pdgfra*^f/f^;*Cnp*-cre) mice, in which *Pdgfra* was specifically ablated in differentiating OLs. Indeed, the expression of Gab1 was significantly increased in the cortex and spinal cord (*Figure 1E*). While these results demonstrated a suppressive effect of PDGF signaling on Gab1

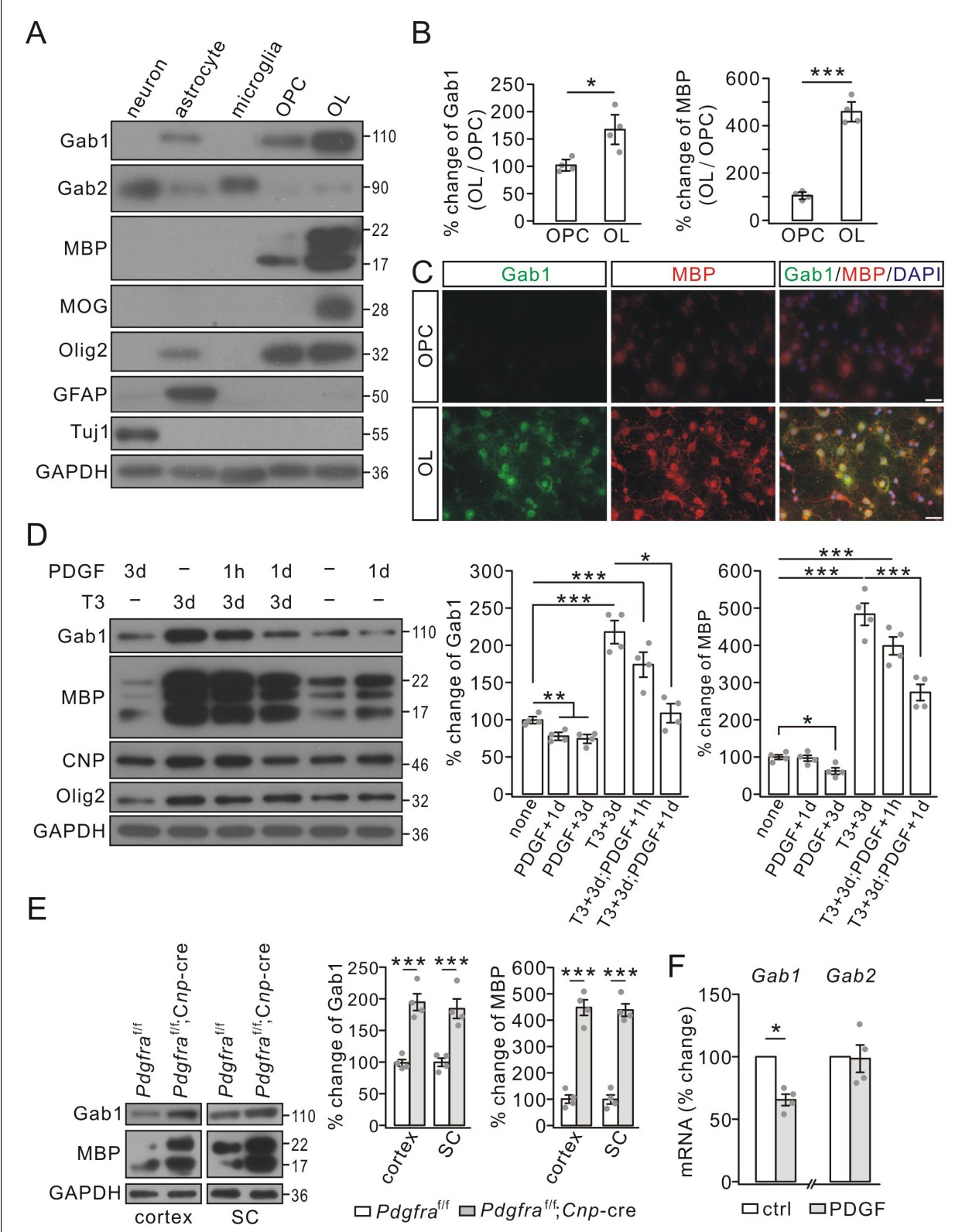

**Figure 1.** Gab1 expression increased during OPC differentiation but was reduced by PDGF in vitro. (**A**) The expressions of Gab1, Gab2, myelin-related proteins, and cell-specific marker proteins in cultured neurons, astrocytes, microglia, OPCs, and OLs. (**B**) The blots of Gab1 and MBP were normalized to corresponding GAPDH and their ratios in OL *vs.* OPC (OL/OPC) were shown as the percentage changes of OPC group. Gab1: 100 ± 7% (OPC) and 169 ± 21% (OL), p=0.014. MBP: 100 ± 10% (OPC) and 459 ± 31% (OL), p=0.00002, n = 4/group, *t*-test, df = *t*(7). (**C**) The immunostaining of Gab1 and

*Figure 1 continued on next page*

*Figure 1 continued*

MBP in cultured OPCs and OLs. Scale bars: 20 μm. (**D**) PDGF and triiodothyronine (**T3**) were administered to OPC cultures as indicated. Gab1 and MBP were normalized to GAPDH and percentage changes are shown in bar graphs. Gab1: 100 ± 4% (none), 78 ± 4% (PDGF+1d), 74 ± 5% (PDGF+3d), 218 ± 15% (T3+3d), 174 ± 16% (T3+3d;PDGF+1 hr), and 108 ± 12% (T3+3d;PDGF+1d), p values: 0.0056 (none *vs* PDGF+1d), 0.0044 (none *vs* PDGF+3d), 0.00015 (none *vs* T3+3d), 0.0021 (none *vs* T3+3d;PDGF+1 hr), and 0.046 (T3+3d;PDGF+1 hr *vs* T3+3d;PDGF+1d). MBP: 100 ± 7% (none), 97 ± 9% (PDGF +1d), 63 ± 10% (PDGF+3d), 484 ± 34% (T3+3d), 399 ± 28% (T3+3d;PDGF+1 hr), and 274 ± 26% (T3+3d;PDGF+1d), p values: 0.012 (none *vs* PDGF+3d), 0.000015 (none *vs* T3+3d), 0.000019 (none *vs* T3+3d;PDGF+1 hr), and 0.0013 (T3+3d *vs* T3+3d;PDGF+1d). $n = 4$/group. ANOVA, $df = F(4, 19)$. (**E**) The expressions of Gab1, MBP and GAPDH in the cerebral cortex and spinal cord (SC) from P21 *Pdgfra*^f/f and *Pdgfra*^f/f;*Cnp*-Cre mice. Gab1 and MBP were normalized to corresponding GAPDH and the percentage changes are shown. Gab1: 100 ± 6% (cortex; *Pdgfra*^f/f) and 195 ± 15% (cortex; *Pdgfra*^f/f;*Cnp*-Cre) (p=0.00053); 100 ± 8% (SC; *Pdgfra*^f/f) and 185 ± 18% (SC; *Pdgfra*^f/f;*Cnp*-Cre) (p=0.0023). MBP: 100 ± 22% (cortex; *Pdgfra*^f/f) and 449 ± 34% (cortex; *Pdgfra*^f/f;*Cnp*-Cre) (p=0.000049); 100 ± 20% (SC; *Pdgfra*^f/f) and 439 ± 27% (SC; *Pdgfra*^f/f;*Cnp*-Cre) (p=0.000024). $n = 4$/group. t-test, $df = t(7)$. (**F**) The mRNA levels of *Gab1* and *Gab2* were quantified by comparative Ct method. The ratios of *Gab*s to *Gapdh* in control (ctrl) and PDGF (1d) groups were calculated and normalized to the control, and the percentage changes are shown in bar graphs. *Gab1*: 66 ± 5% (p=0.032 *vs* control), $n = 4$/group. *Gab2*: 99 ± 14% (p=0.75 *vs* control), $n = 4$/group, t-test, $df = t(7)$. Gray dots indicate individual data points. *p<0.05. ***p<0.001.

expression, a remaining question was how PDGF signaling negatively regulates Gab1. We measured the mRNA levels of *Gab1* and *Gab2* in cultured OPCs treated with PDGF-AA. Our results showed that *Gab1* mRNA was reduced after 1 day treatment with PDGF-AA, whereas *Gab2* mRNA was not altered (*Figure 1F*), implying that PDGF signaling affects *Gab1* transcription.

## Gab1 is specifically regulated by PDGF signaling

As an adaptor molecule, Gab1 is suggested to interact with a number of growth factors in neural progenitor cells (*Korhonen et al., 1999*; *Cai et al., 2002*; *Mao and Lee, 2005*). Our next question was whether the regulation of Gab1 in OLs is controlled by other growth factors besides PDGF. Therefore, we administered EGF (10 ng/ml), insulin-like growth factor-1 (IGF-1, 10 ng/ml), NRG-1 (50 ng/ml), and PDGF (10 ng/ml) individually to OPC cultures for 1 day prior to 3-day treatment with tri-iodothyronine. Our results showed that only PDGF was able to decrease Gab1 expression augmented by triiodothyronine, whereas EGF, NRG-1 and IGF-1 had no effect (*Figure 2A*), suggesting that Gab1 is specifically regulated by PDGF.

We next compared the regulatory effects of PDGF on Gab proteins in cultured OPCs and astrocytes, since Gab1 and Gab2 appeared to be expressed more or less in both cell types (*Figure 1A*). To do so, we administered PDGF or triiodothyronine to cultured OPCs and astrocytes, which could be distinguished by the specific molecular markers (CNP, PDGFRα, and glial fibrillary acidic protein, GFAP). Because the shaking could not completely exclude astrocytes from OPCs and vice versa, it was not surprising that weak bands of GFAP and PDGFRα were shown in OPCs and astrocytes, respectively (*Figure 2B*). PDGF and triiodothyronine again induced opposite effects on Gab1 expression in OPCs, but not in astrocytes (*Figure 2B*). In contrast, Gab2 expression was not affected by PDGF in both astrocytes and OPCs. These results suggest that the expression of Gab1 is selectively regulated by PDGF signaling in OPCs.

## Deletion of *Gab1* in OLs impairs myelination in CNS

Since Gab1 expression was elevated in mature OLs, we investigated its role in CNS myelination using *Gab1* conditional knockout mice generated by mating *Gab1*^f/f mice with various cre transgenic lines, *Olig1*-cre, *Cspg2*-cre, *Campk2a*-cre and *Nestin*-cre. It is acknowledged that the conditional deletion mediated by *Olig1*-cre or *Cspg2*-cre mainly affects OL lineage, whereas conditional deletion mediated by *Nestin*-cre or *Campk2a*-cre affects neural stem cells or excitatory neurons. Western blots revealed that Gab1 was richly expressed in the cerebral cortex derived from *Gab1*^f/f and *Gab1*^f/f;*Campk2a*-cre mice at P21 (*Figure 3A*). However, Gab1 was absent in the cortex from *Gab1*^f/f;*Nestin*-cre, *Gab1*^f/f;*Olig1*-cre and *Gab1*^f/f;*Cspg2*-cre mice (*Figure 3A*). *Gab2* mutation had no additional effects on Gab1 expression in various double mutant mice (*Figure 3A*). It should be noted that, compared with in vitro results that might be affected by culture purity (*Figure 1A*), in vivo data from *Gab1*^f/f;*Olig1*-cre conditional mutants provide convincing evidence showing that Gab1 is solely expressed in OLs.

We chose *Gab1*^f/+;*Olig1*-cre mice as the control to investigate the actions of Gab1 on myelination, which allowed us to minimize side effects of *Olig1*-cre insertion. Our TEM studies showed

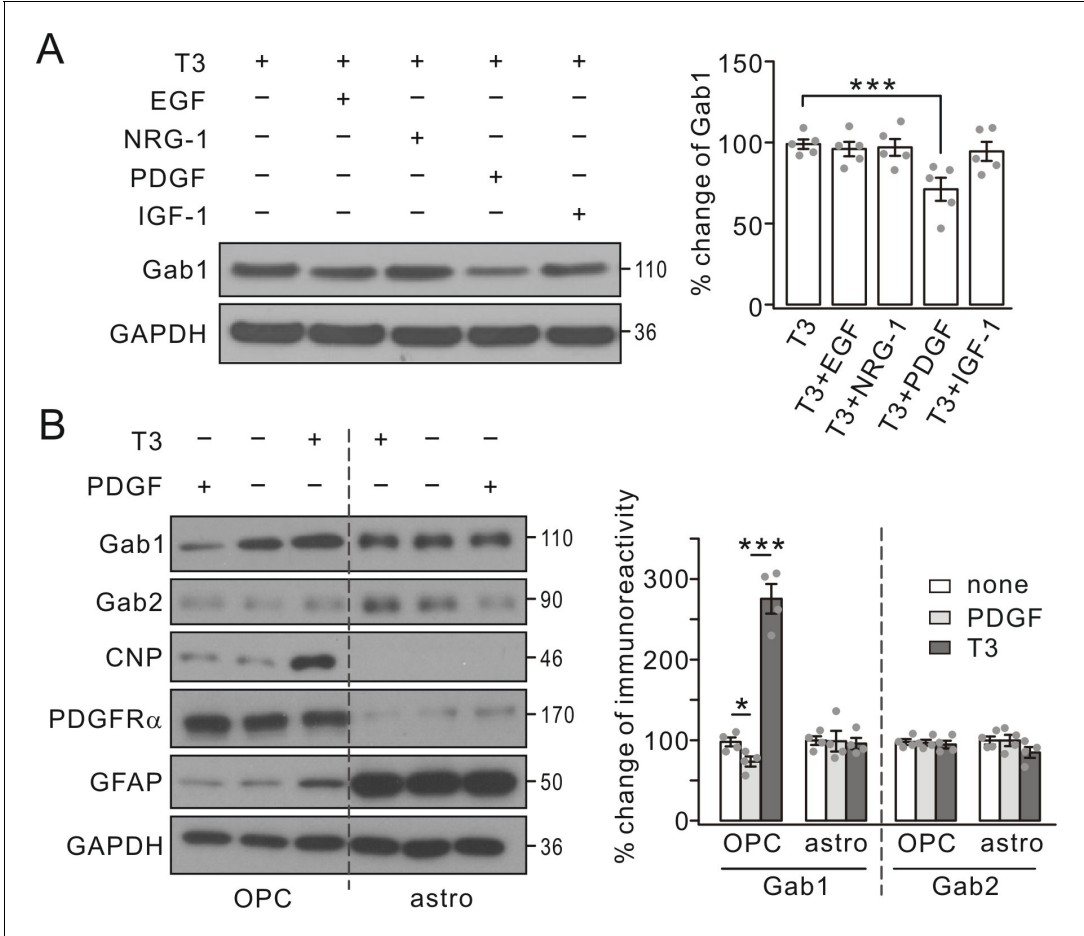

**Figure 2.** Gab1 expression was specifically suppressed by PDGF. (A) Triiodothyronine (T3) was administered to OPC cultures along with EGF, NRG-1, PDGF or IGF-1, as indicated by '+' and '-'. Gab1 expression was normalized to GAPDH and percentage changes are shown in bar graphs. T3: 100 ± 3%. T3+EGF: 97 ± 5%. T3+NRG-1: 98 ± 6%. T3+PDGF: 71 ± 8%. T3+IGF-1: 94 ± 7%. p values: 0.59 (T3 *vs* T3+EGF), 0.75 (T3 *vs* T3+NRG-1), 0.0072 (T3 *vs* T3 +PDGF), and 0.52 (T3 *vs* T3+IGF-1). n = 5/group. ANOVA, df = $F(4, 20)$. (B) T3 or PDGF was administered to OPC and astrocytic (astro) cultures, as indicated by '+' and '-'. Lysates were probed with antibodies to proteins labeled in the left. The expression of Gab1 and Gab2 was normalized to corresponding GAPDH and the percentage changes are shown in bar graphs. Gab1-OPC: 100 ± 5% (none), 74 ± 7% (PDGF), 276 ± 21% (T3), and p values: 0.029 (none *vs* PDGF) and 0.000087 (none *vs* T3). Gab1-astro: 100 ± 6% (none), 100 ± 15% (PDGF), 96 ± 8% (T3), and p values: 0.96 (none *vs* PDGF) and 0.69 (none *vs* T3). Gab2-OPC: 100 ± 4% (none), 97 ± 4% (PDGF), 95 ± 5% (T3), and p values: 0.85 (none *vs* PDGF) and 0.61 (none *vs* T3). Gab2-astro: 100 ± 6% (none), 100 ± 8% (PDGF), 85 ± 8% (T3), and p values: 0.99 (none *vs* PDGF) and 0.13 (none *vs* T3). n = 4/group. ANOVA, df = $F(2, 9)$. Gray dots indicate individual data points. ***p<0.001.

significantly fewer myelinated axons in *Gab1*[f/f];*Olig1*-cre mice than in *Gab1*[f/+];*Olig1*-cre mice at P21 (*Figure 3B*). In *Gab1*[f/+];*Olig1*-cre mice, 55% of axons were myelinated, whereas only 34% of axons were myelinated in *Gab1*[f/f];*Olig1*-cre mice (*Figure 3B'*). Also in *Gab1*[f/f];*Olig1*-cre mice, the proportion of small-diameter axons (<1.0 μm) was increased by 100%, whereas the large-diameter axons (1.0–2.0 μm) decreased by 39% (*Figure 3B''*), suggesting that axon diameter might be reduced by conditional *Gab1* deletion. Meanwhile, myelin thickness was unaltered in *Gab1*[f/f];*Olig1*-cre littermates, as indicated by an unchanged g-ratio of myelin sheaths (*Figure 3C*). These results suggest a hypomyelination phenotype in *Gab1*[f/f];*Olig1*-cre mice, which was further confirmed by immunohistochemical analyses. MBP staining revealed a broad loss of MBP-positive fibers in the cerebral cortex, the corpus callosum, the hippocampus, and the cerebellum of *Gab1*[f/f];*Olig1*-cre mice (*Figure 3D*). The black-gold myelin staining also showed that *Gab1*[f/f];*Olig1*-cre and *Gab1*[f/f];*Cspg2*-cre mice (P21) had a remarkable reduction of white matter tracts in the corpus callosum compared to *Gab1*[f/+]; *Olig1*-cre or *Gab1*[f/f] control mice (*Figure 3E*). To determine whether the hypomyelination phenotype

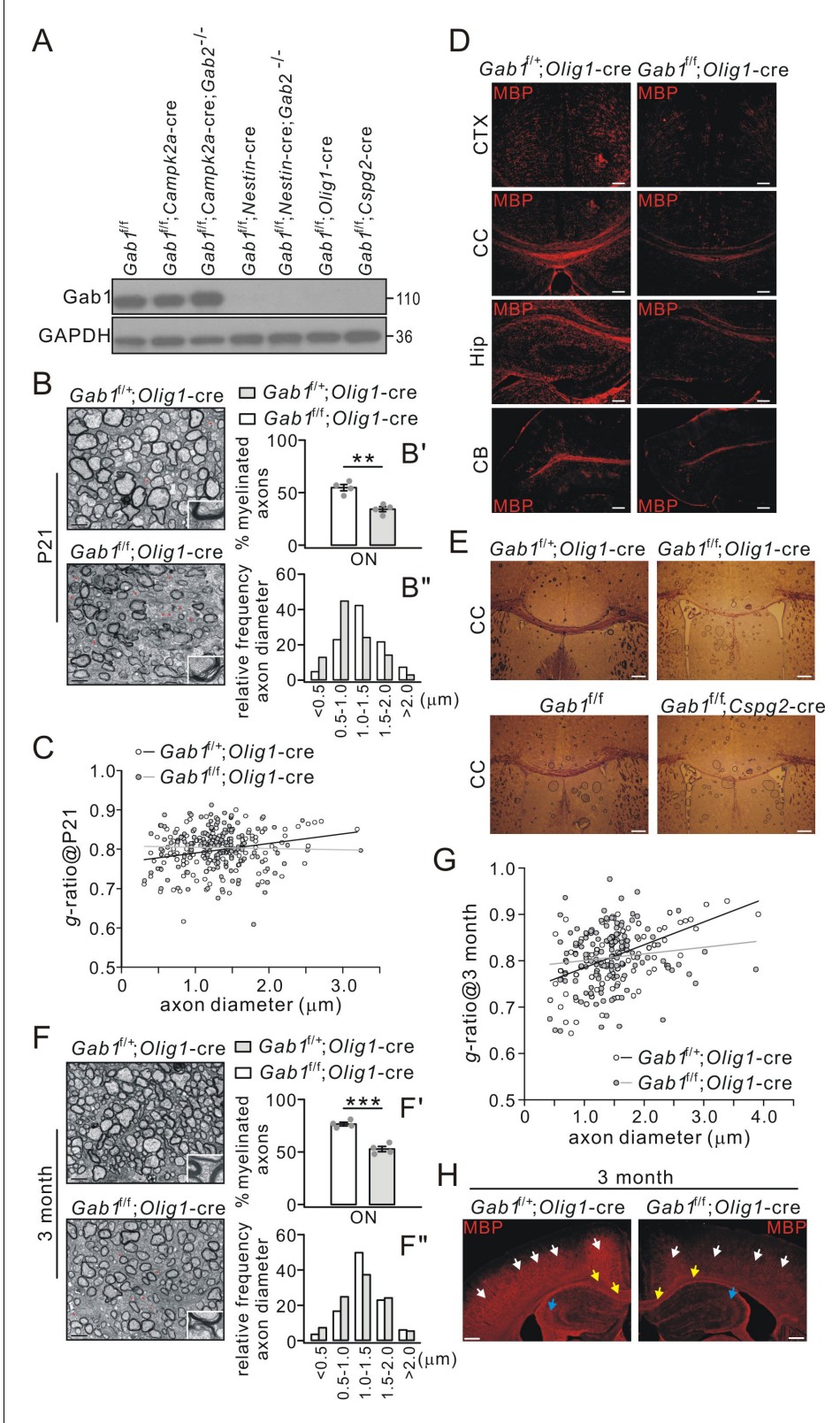

**Figure 3.** Impaired CNS myelination in conditional *Gab1*-knockout mice. (**A**) Gab1 expression in *Gab1*^f/f^, *Gab1* conditional knockout, and *Gab1/Gab2* double mutant mice (**P21**). (**B**) TEM images of optic nerve (ON) from *Gab1*^f/+^;*Olig1*-cre and *Gab1*^f/f^;*Olig1*-cre mice (**P21**). Unmyelinated axons are indicated by red asterisks. Scale bars, 0.5 μm. The insets show typical axons from two groups. (**B'**) shows the percentages of myelinated axons: *Gab1*^f/+^; *Olig1*-cre: 55 ± 3% (4 animals); *Gab1*^f/f^;*Olig1*-cre: 34 ± 3% (4 animals), p=0.002, *t*-test, df = *t*(7). (**B"**) shows the distribution of axonal size in optic nerve

*Figure 3 continued on next page*

*Figure 3 continued*

(*n* = 160 axons in each group). (**C**) The relationship between diameters and *g*-ratios of axons from *Gab1*<sup>f/+</sup>;*Olig1*-cre and *Gab1*<sup>f/f</sup>;*Olig1*-cre mice (**P21**). Averaged *g*-ratios were 0.80 ± 0.04 (*Gab1*<sup>f/+</sup>;*Olig1*-cre, 154 axons from four animals) and 0.80 ± 0.05 (*Gab1*<sup>f/f</sup>;*Olig1*-cre, 148 axons from four animals), p=0.26, *t*-test, df = *t*(300). (**D**) MBP staining of cerebral cortex (CTX), corpus callosum (CC), hippocampus (Hip), and cerebellum (CB) from *Gab1*<sup>f/+</sup>;*Olig1*-cre and *Gab1*<sup>f/f</sup>;*Olig1*-cre mice (**P21**). Scale bars, 50 μm. (**E**) Black-gold staining of corpus callosum from *Gab1*<sup>f/+</sup>;*Olig1*-cre *vs.* *Gab1*<sup>f/f</sup>;*Olig1*-cre mice (**P21**) or from *Gab1*<sup>f/f</sup> *vs.* *Gab1*<sup>f/f</sup>;*Cspg2*-cre mice (**P21**). Scale bars, 200 μm. (**F**) TEM images of optic nerve (ON) from *Gab1*<sup>f/+</sup>;*Olig1*-cre and *Gab1*<sup>f/f</sup>;*Olig1*-cre mice at 3 month. Unmyelinated axons are indicated by red asterisks. Scale bars, 0.5 μm. The insets show typical axons from two groups. (**F'**) shows the percentages of myelinated axons: *Gab1*<sup>f/+</sup>;*Olig1*-cre: 77 ± 2% (4 animals); *Gab1*<sup>f/f</sup>;*Olig1*-cre: 53 ± 3% (4 animals), p=0.0002, *t*-test, df = *t*(7). (**F''**) shows the distribution of axonal size in optic nerve (*n* = 160 axons in each group). (**G**) The relationship between diameters and *g*-ratios of axons from mice at 3 months. Averaged *g*-ratios were 0.81 ± 0.06 (*Gab1*<sup>f/+</sup>;*Olig1*-cre, 141 axons from four animals) and 0.81 ± 0.06 (*Gab1*<sup>f/f</sup>;*Olig1*-cre, 141 axons from four animals), p=0.87, *t*-test, df = *t*(280). (**H**) MBP staining in the cerebral cortex (white arrows), the corpus callosum (yellow arrows) and the hippocampus (blue arrows) from *Gab1*<sup>f/+</sup>;*Olig1*-cre and *Gab1*<sup>f/f</sup>;*Olig1*-cre mice at 3 months. Note that MBP intensity was reduced in *Gab1*<sup>f/f</sup>;*Olig1*-cre compared to *Gab1*<sup>f/+</sup>;*Olig1*-cre mice. Scale bars, 200 μm. Gray dots indicate individual data points. \*\*p<0.01. \*\*\*p<0.001.

persists at the later developmental stage, we examined the myelin structure in 3-month-old *Gab1*<sup>f/+</sup>;*Olig1*-cre and *Gab1*<sup>f/f</sup>;*Olig1*-cre mice. TEM experiment showed that, similar to the age of P21, significantly fewer myelinated axons were found in *Gab1*<sup>f/f</sup>;*Olig1*-cre mice at the age of 3 months (*Figure 3F and F'*), meanwhile myelin thickness was unaltered indicated by unchanged *g*-ratio of myelin sheath (*Figure 3G*). Likewise, the proportion of small-diameter axons increased but the large-diameter axons decreased in *Gab1*<sup>f/f</sup>;*Olig1*-cre mice at this age (*Figure 3F''*). MBP-positive fibers were also significantly fewer in the cerebral cortex, the corpus callosum and the hippocampus of 3-month-old *Gab1*<sup>f/f</sup>;*Olig1*-cre mice (*Figure 3H*). These data support the important roles of Gab1 in CNS myelination.

Myelin-specific proteins appear in OLs prior to the onset of myelination and are continually produced by OLs during the anabolism and catabolism of myelin sheath (*Sternberger et al., 1978*). Myelin proteins are not only the major components of myelin but also the characteristic indicators of myelination capacity. Thus, myelin-specific proteins were examined to define the effects of *Gab1*-knockout on myelination. Our results showed that the expression of MBP, CNP and MOG was attenuated in the cerebral cortex, the hippocampus, the cerebellum, spinal cord, corpus callosum, and optic nerves from *Gab1*<sup>f/f</sup>;*Olig1*-cre and *Gab1*<sup>f/f</sup>;*Cspg2*-cre mice at P21 compared to those from *Gab1*<sup>f/+</sup>;*Olig1*-cre and *Gab1*<sup>f/f</sup> controls (*Figure 4A and B*). The myelin-related proteins were also examined in 3-month-old *Gab1*<sup>f/+</sup>;*Olig1*-cre and *Gab1*<sup>f/f</sup>;*Olig1*-cre mice. Similar to the mice at P21, the expression of MBP, CNP and MOG significantly decreased in the cerebral cortex and the corpus callosum of *Gab1*<sup>f/f</sup>;*Olig1*-cre mice compared to *Gab1*<sup>f/+</sup>;*Olig1*-cre mice at this age (*Figure 4—figure supplement 1*). To examine the difference in myelin components, myelin fractions were isolated and purified from the brain and myelin-specific proteins were examined (*Saher et al., 2005*). Similar to their total expressions, MBP, PLP and MOG in myelin fractions also remarkably decreased in *Gab1*<sup>f/f</sup>;*Olig1*-cre mice at P21 (*Figure 4C*).

Taken together, our TEM, immunohistochemistry, and protein assay demonstrate that *Gab1* deletion in OLs leads to myelin deficits in the CNS. We compared the expression of myelin-specific proteins in *Gab2*-knockout mice as well, and our results showed no difference in the expression of MBP, CNP and MOG proteins between control and *Gab2*<sup>-/-</sup> mice (*Figure 4D*).

## *Gab1* ablation reduces OPC differentiation

To investigate how *Gab1* deficiency causes the hypomyelination, we examined the expression of specific cellular markers of OL lineage cells (Olig2), OPCs (PDGFRα) and OLs (CC1) in *Gab1*<sup>f/+</sup>;*Olig1*-cre and *Gab1*<sup>f/f</sup>;*Olig1*-cre mice. We found that the density of Olig2+ OLs was lower in the cerebral cortex and the corpus callosum of *Gab1*<sup>f/f</sup>;*Olig1*-cre mice than that in *Gab1*<sup>f/+</sup>;*Olig1*-cre mice at P21 (*Figure 5A, B and C*). We continued to examine the differentiation and proliferating capacity of OPCs using double staining of Olig2/CC1 or Olig2/PDGFRα in the cerebral cortex and the corpus callosum. Our results showed that the density of differentiated OLs (positive to Olig2 and CC1) decreased by 47% in *Gab1*<sup>f/f</sup>;*Olig1*-cre mice compared with *Gab1*<sup>f/+</sup>;*Olig1*-cre mice (*Figure 5A and C*). By contrast, Olig2/PDGFRα double staining in the cerebral cortex and the corpus callosum showed no difference in the density of PDGFRα+Olig2+ OPCs between *Gab1*<sup>f/f</sup>;*Olig1*-cre and *Gab1*<sup>f/+</sup>;*Olig1*-cre littermates (*Figure 5B and C*). The reduced OL phenotype was also found in the

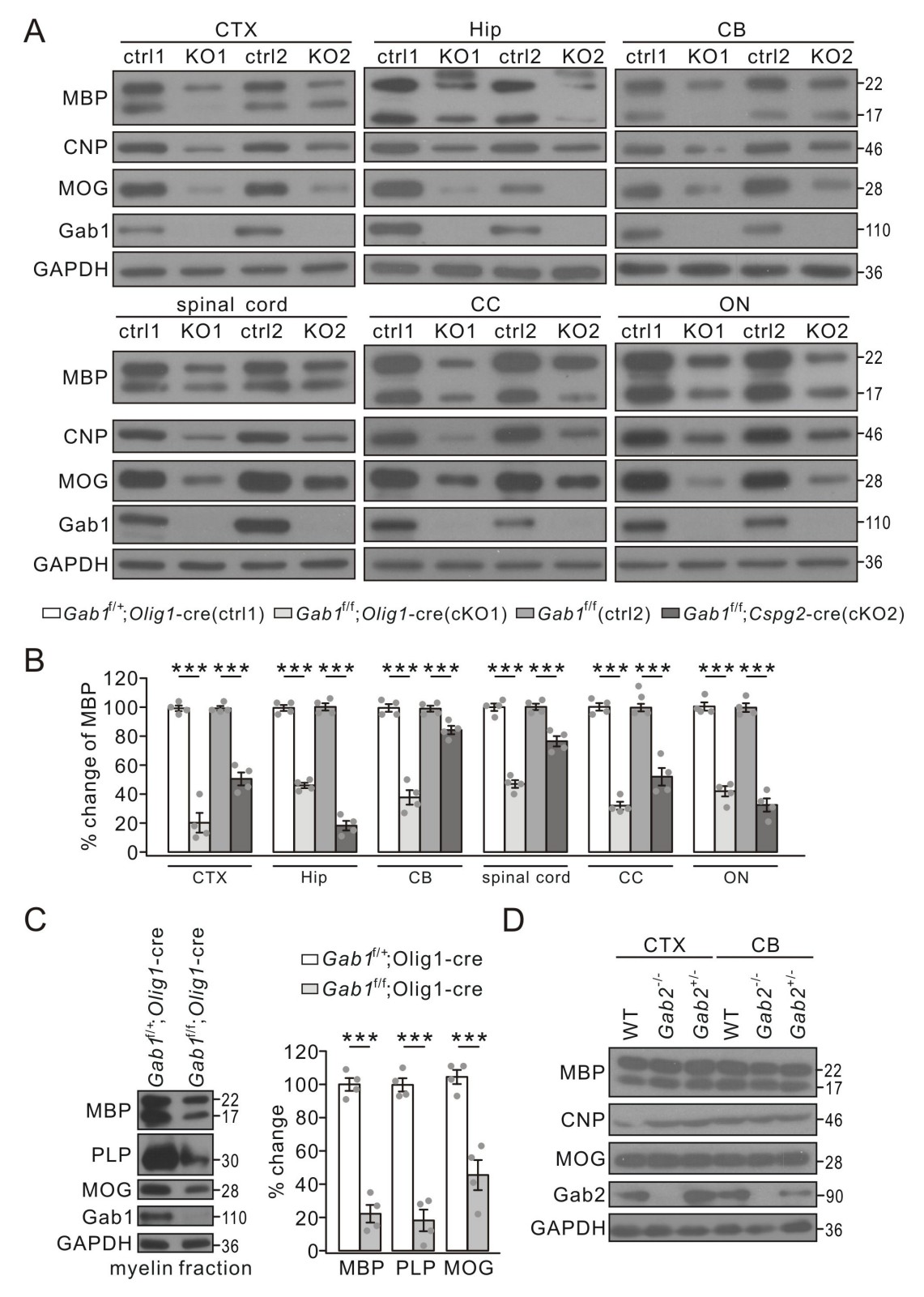

**Figure 4.** The reduced expression of myelin-related proteins in conditional *Gab1* knockout mice. (**A**) Western blots of myelin proteins in the cerebral cortex (CTX), the hippocampus (Hip), the cerebellum (CB), spinal cord, the corpus callosum (CC), and optic nerve (ON) from *Gab1*^f/+;*Olig1*-cre (ctrl1) *vs.* *Gab1*^f/f;*Olig1*-cre (KO1) mice or *Gab1*^f/f (ctrl2) *vs.* *Gab1*^f/f;*Cspg2*-cre (KO2) mice at P21. (**B**) MBP expression was normalized to GAPDH and percentage changes are shown. In the order of ctrl1, KO1, ctrl2, and KO2, changes are: CTX, 100 ± 2% and 20 ± 8% (p=0.000030), 100 ± 2% and 51 ± 5%

*Figure 4 continued on next page*

Figure 4 continued

(p=0.000051); Hip, 100 ± 2% and 46 ± 2% (p=0.000001), 100 ± 3% and 18 ± 4% (p=0.000001); CB, 100 ± 3% and 38 ± 6% (p=0.000034), 100 ± 2% and 84 ± 3%(p=0.0058); spinal cord, 100 ± 3% and 47 ± 3% (p=0.000007), 100 ± 3% and 76 ± 4% (p=0.0013); CC, 100 ± 3% and 32 ± 3% (p=0.000002), 100 ± 3% and 52 ± 7%(p=0.00036); ON, 100 ± 3% and 42 ± 4% (p=0.000014), 100 ± 3% and 33 ± 5% (p=0.000017). $n$ = 4/group. $t$-test, df = $t$(7). (**C**) Myelin proteins in myelin fractions of corpus callosum from $Gab1^{f/+}$;$Olig1$-cre and $Gab1^{f/f}$;$Olig1$-cre mice at P60. GAPDH was internal control. Expressions of MBP, PLP and MOG were normalized to corresponding GAPDH and the percentage changes are shown. MBP: 100 ± 4% ($Gab1^{f/+}$;$Olig1$-cre) and 23 ± 6% ($Gab1^{f/f}$; $Olig1$-cre) (p=0.000023). PLP: 100 ± 4% ($Gab1^{f/+}$;$Olig1$-cre) and 18 ± 7% ($Gab1^{f/f}$;$Olig1$-cre) (p=0.000039). MOG: 100 ± 5% ($Gab1^{f/+}$;$Olig1$-cre) and 45 ± 10% ($Gab1^{f/f}$;$Olig1$-cre) (p=0.0011). $n$ = 4/group, $t$-test, df = $t$(7). (**D**) Western blots of myelin proteins in the cerebral cortex (CTX) and the cerebellum (CB) from WT, $Gab2^{-/-}$ and $Gab2^{+/-}$ mice at P21. The experiment was repeated four times. Gray dots indicate individual data points. ***p<0.01.

The online version of this article includes the following figure supplement(s) for figure 4:

**Figure supplement 1.** The reduced expression of myelin-related proteins in 3-month-old $Gab1^{f/f}$;$Olig1$-cre mice.

late developmental of adult mice. In the cerebral cortex and the corpus callosum of $Gab1^{f/f}$;$Olig1$-cre at the age of 3 months, our immunohistochemical experiment showed that the densities of both Olig2+ and Olig2+CC1+ OLs were much reduced compared to $Gab1^{f/+}$;$Olig1$-cre mice (**Figure 5—figure supplement 1**).

Although these results suggest that conditional deletion of $Gab1$ impairs OPC differentiation but not OPC proliferation, it is not answered whether Gab1 functions in mature OLs as well. To address this question, we examined myelin organization in $Gab1^{f/+}$;$Plp1$-creER and $Gab1^{f/f}$;$Plp1$-creER mice, where Gab1 was ablated in mature OLs of $Gab1^{f/f}$;$Plp1$-creER mice following tamoxifen treatment. Eriochrome cyanine staining showed that tamoxifen did not change the density of white matter tracts in the corpus callosum in both types of mice 4 weeks after tamoxifen injection (**Figure 5D**), meanwhile western blotting assay showed that $Gab1$ was deleted in the corpus callosum of $Gab1^{f/f}$; $Plp1$-creER mice treated with tamoxifen while no change found in other conditions (**Figure 5E**). Although these results suggest that Gab1 does not participate in maintaining mature OLs, it must be noted that there is a possibility that tamoxifen causes a different phenotype in longer durations, for example, the long-time effects of tamoxifen induction (**Koenning et al., 2012**) or Olig1 deletion (**Arnett et al., 2004**) on mature OLs.

The action of Gab1 on OPC differentiation was further confirmed by $Gab1$-knockdown experiments using lentiviral transfection of GFP-tagged $Gab1$ shRNA in cultured OPCs. In order to compare the levels of maturation between naive control and $Gab1$ shRNA groups, OPCs were grown in the same density in two groups, as indicated by Olig2 staining (**Figure 6A**). This strategy allowed us to focus on OPC differentiation without considering the proliferation. In the control group, triiodothyronine treatment yielded a large number of MBP+ cells (**Figure 6A**). In contrast, $Gab1$ down-regulation by shRNA resulted in markedly fewer MBP+ cells after triiodothyronine treatment (**Figure 6A**). Furthermore, MBP protein was dramatically decreased by $Gab1$ shRNA, which was verified by western blots (**Figure 6B**). Hence, in vitro evidence supports the conclusion that Gab1 regulates the differentiation of OPCs.

## Gab1 binds to GSK3β and modulates its activity

Although the results above show that $Gab1$ deficiency interrupts OPC differentiation and CNS myelination, the downstream effectors of Gab1 are not understood. It occurred to us that GSK3β and β-catenin signaling is critical for proper CNS myelination (**Azim and Butt, 2011**; **Zhou et al., 2014**), but their upstream factor is unclear. Thus, an interesting question was whether Gab1 affects GSK3β and β-catenin in OLs. We explored this possibility by assessing the binding capacity between Gab1 and GSK3β. Interestingly, in vivo Co-IP in cortical tissues showed that GSK3β was robustly precipitated by Gab1 and vice versa (**Figure 7A**). Akt1, a signaling hub of growth factors in many biological processes and an upstream regulator of GSK3β, bound to Gab1 as well (**Figure 7A**). Moreover, both GSK3β and PDGFRα were precipitated by Gab1 in OPCs and OLs in vitro (**Figure 7B**), confirming the binding between Gab1 and GSK3β. These results indicate that Gab1 is a mediator between PDGFRα and GSK3β.

It was previously reported that GSK3β phosphorylation at S9 correlates with the differentiation of OPCs and myelin gene expression (**Kim et al., 2009**; **Zhou et al., 2014**). The robust binding of GSK3β and Akt1 to Gab1 raises a possibility that Gab1 may modulate the activity of GSK3β and Akt1. It is known that phosphorylated GSK3β and Akt are their activated forms: the phosphorylation

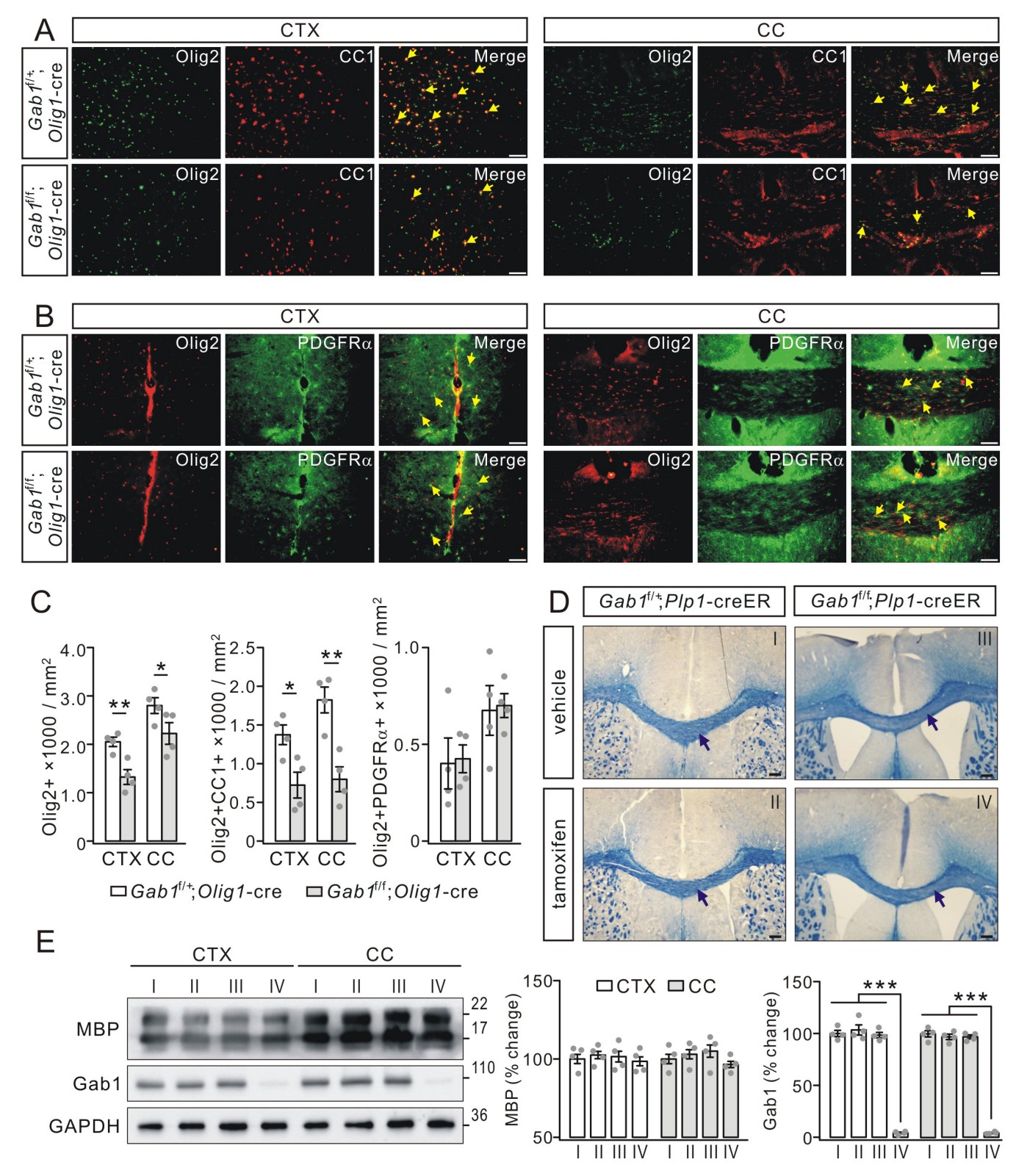

**Figure 5.** *Gab1* deletion impairs OPCs differentiation. (**A**) The double-immunostaining of Olig2 and CC1 in the cerebral cortex (CTX) and corpus callosum (CC) from *Gab1*[f/+];*Olig1*-cre and *Gab1*[f/f];*Olig1*-cre mice at P21. Cells positive to Olig2 and CC1 were recognized as myelinated OLs (yellow arrows). Scale bars, 50 μm. (**B**) The double-immunostaining of Olig2 and PDGFRα in the cerebral cortex and corpus callosum from *Gab1*[f/+];*Olig1*-cre and *Gab1*[f/f];*Olig1*-cre mice at P21. Cells positive to Olig2 and PDGFRα were recognized as OPCs (yellow arrows). Scale bars, 50 μm. (**C**) Bar graphs

*Figure 5 continued on next page*

*Figure 5 continued*

show the densities of Olig2+, Olig2+CC1+, and Olig2+PDGFRα+ cells in $Gab1^{f/f}$;$Olig1$-cre and $Gab1^{f/+}$;$Olig1$-cre mice at P21. The densities of Olig2+ cells were 2050 ± 111 ($Gab1^{f/+}$;$Olig1$-cre) and 1325 ± 176 cells/mm$^2$ ($Gab1^{f/f}$;$Olig1$-cre) (CTX; p=0.007), and 2800 ± 188 ($Gab1^{f/+}$;$Olig1$-cre) and 2225 ± 258 cells/mm$^2$ ($Gab1^{f/f}$;$Olig1$-cre) (CC; p=0.024). The densities of Olig2+CC1+ cells were 1375 ± 148 ($Gab1^{f/+}$;$Olig1$-cre) and 724 ± 193 cells/mm$^2$ ($Gab1^{f/f}$;$Olig1$-cre) (CTX; p=0.02), and 1824 ± 193 ($Gab1^{f/+}$;$Olig1$-cre) and 800 ± 186 cells/mm$^2$ ($Gab1^{f/f}$;$Olig1$-cre) (CC; p=0.005). The densities of Olig2+PDGFRα+ cells were 401 ± 152 ($Gab1^{f/+}$;$Olig1$-cre) and 425 ± 83 cells/mm$^2$ ($Gab1^{f/f}$;$Olig1$-cre) (CTX; p=0.87), and 675 ± 148 ($Gab1^{f/+}$;$Olig1$-cre) and 700 ± 71 cells/mm$^2$ ($Gab1^{f/f}$;$Olig1$-cre) (CC; p=0.87). n = 4/group, t-test, df = t(7). (D) $Gab1^{f/+}$;$Plp1$-creER and $Gab1^{f/f}$;$Plp1$-creER mice were (P180) treated with either vehicle or tamoxifen, and the corpus callosum of mice were collected at day 28 p.i. and stained with cyanine. No difference in the density of white matter tracts between vehicle and tamoxifen groups, as indicated by black arrows. Scale bars, 100 μm. (E) Western blots show that Gab1 was deleted in the cerebral cortex and the corpus callosum of $Gab1^{f/f}$;$Plp1$-creER mice treated with tamoxifen. The Roman numbers I, II, III and IV that are also marked in (D) represent individual condition, $Gab1^{f/+}$;$Plp1$-creER+vehicle, $Gab1^{f/+}$;$Plp1$-creER+tamoxifen, $Gab1^{f/f}$;$Plp1$-creER+vehicle, and $Gab1^{f/f}$;$Plp1$-creER+tamoxifen, The percentage changes of MBP expression normalized to condition I were: cortex, 100 ± 4% (I), 102 ± 3%(II), 102 ± 5% (III), 99 ± 4% (IV); corpus callosum: 100 ± 4% (I), 103 ± 4%(II), 105 ± 6% (III), 97 ± 3% (IV). The percentage changes of Gab1 expression normalized to condition I were: cortex, 100 ± 3% (I), 103 ± 6% (II), 98 ± 3% (III), 4 ± 1% (IV); corpus callosum, 100 ± 4% (I), 97 ± 3% (II), 97 ± 2% (III), 4 ± 1% (IV). For all statistics, n = 4/group, t-test, df = t(7). Gray dots indicate individual data points. *p<0.05. **p<0.01. ***p<0.001.

The online version of this article includes the following figure supplement(s) for figure 5:

**Figure supplement 1.** Mature OLs decreased in $Gab1^{f/f}$;$Olig1$-cre mice at 3 months.

at S9 is a negative regulator and the phosphorylation at Y216 is a positive regulator of GSK3β activity (*Wang et al., 1994*; *Hughes et al., 1993*); Akt1 is phosphorylated at S473 and T308, and the phosphorylation at these two sites is necessary and sufficient for Akt1 activation (*Alessi et al., 1997*; *Sarbassov et al., 2005*). Thus, we assessed the phosphorylation of GSK3β and Akt1 in $Gab1^{f/+}$; $Olig1$-cre and $Gab1^{f/f}$;$Olig1$-cre mice at P21. Our results demonstrated that only GSK3β-S9 phosphorylation was increased by the ablation of Gab1 in OLs, while the phosphorylations of GSK3β-Y216, Akt1-S473, and Akt1-T308 as well as the total expression of GSK3β, Akt1, and β-catenin was unaltered (*Figure 7C*). Thus, the changed phosphorylation of GSK3β-S9 may explain how Gab1 controls OPC differentiation according to previous work (*Azim and Butt, 2011*; *Zhou et al., 2014*). To examine what happens in myelin components, we evaluated the phosphorylation of GSK3β-S9 in myelin fractions isolated from the brain of $Gab1^{f/+}$;$Olig1$-cre and $Gab1^{f/f}$;$Olig1$-cre mice (P60). Likewise, we found that GSK3β-S9 phosphorylation increased while its expression did not, whereas β-catenin expression in the myelin fraction slightly increased (*Figure 7D*).

Once again, the regulation of GSK3β-S9 phosphorylation by Gab1 was investigated in cultured OPCs infected with Gab1 shRNA lenti-virus. In consistent with control group, GSK3β-S9 phosphorylation significantly increased in Gab1 shRNA group, whereas GSK3β-Y216 phosphorylation was unchanged (*Figure 8A*).

## Gab1 controls β-catenin nuclear accumulation and expression of transcription factors

Activated GSK3β causes the degradation of β-catenin (*Aberle et al., 1997*), which participates in the development of OLs (*Fancy et al., 2009*). Our previous work also shows that GSK3β inhibition promotes the nuclear accumulation of β-catenin in OPCs (*Zhou et al., 2014*). Since GSK3β activity was decreased by Gab1 ablation (*Figure 7C*), we investigated whether conditional knockout of Gab1 changes the nuclear accumulation of β-catenin. As shown in *Figure 9A*, nuclear β-catenin significantly increased in $Gab1^{f/f}$;$Olig1$-cre mice P21 while its total was unchanged. These results affirm that GSK3β controls OPC differentiation through regulating the nuclear accumulation of β-catenin.

GSK3β regulates transcription factors required for the expression of myelin proteins (*Zhou et al., 2014*). We speculated that conditional knockout of Gab1 might exert similar effects on the transcription of myelin-related genes. Indeed, the mRNA levels of MBP, PLP and MOG were significantly reduced in $Gab1^{f/f}$;$Olig1$-cre mice compared to $Gab1^{f/+}$;$Olig1$-cre mice at P21 (*Figure 9B*). Furthermore, the conditional deletion of Gab1 down-regulated a number of positive factors for the transcription of myelin-related genes, including Sox10, Olig2, and zinc finger protein YY1, in the cytoplasm and nucleus, while increasing the nuclear expression of Sox6, a repressor protein (*Figure 9C*). The regulation of transcription factors by Gab1 was examined in cultured OPCs as well. We found that Gab1 shRNA significantly decreased the expressions of Sox10, Olig2 and myelin regulatory factor (Mrf), but increased the expression of Sox6 (*Figure 8B*), consistent with in vivo results.

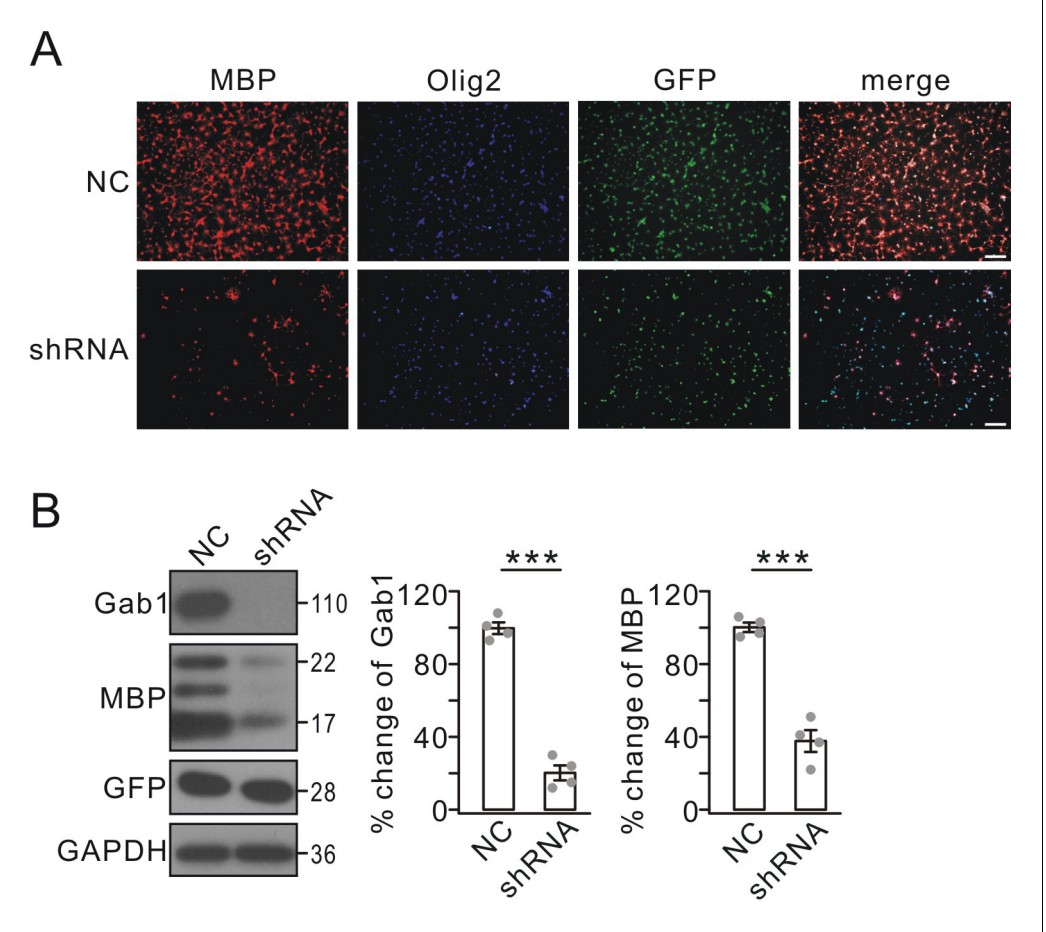

**Figure 6.** Effects of *Gab1* knockdown on OPC differentiation in vitro. (**A**) OPC cultures were transfected with lentiviral vector-encoding with GFP (NC) or GFP-tagged *Gab1* shRNA (shRNA), stimulated with T3 for 3 days, and immunostained with anti-MBP (red) and anti-Olig2 (blue) antibodies. Scale bar, 40 μm. (**B**) Expressions of Gab1, MBP, GFP, and GAPDH in OPCs treated with NC or shRNA following triiodothyronine treatment. Gab1 and MBP were normalized to corresponding GAPDH and percent changes are summarized. Gab1: $100 \pm 4\%$ (NC) and $20 \pm 5\%$ (shRNA) (p=0.00005). MBP: $100 \pm 3\%$ (NC) and $37 \pm 7\%$ (shRNA) (p=0.000075). $n$ = 4/group. *t*-test, df = *t*(7). Gray dots indicate individual data points. \*\*\*p<0.001.

Since these factors are required for the transcription of myelin-related genes (*Fu et al., 2002*; *Stolt et al., 2002*; *He et al., 2007*; *Emery et al., 2009*), the effects of Gab1 on OPC differentiation may be mediated by them.

## Discussion

In the present work, we revealed previously unidentified roles of Gab1: it is a downstream effector of PDGF signaling and promotes OL differentiation by modulating the activity of GSK3β and β-catenin. The functions of Gab1 in the mitotic processes of neural progenitor cells have been reported (*Cai et al., 2002*; *Korhonen et al., 1999*; *Mao and Lee, 2005*), but this is the first report regarding the functions of Gab1 in OL development and CNS myelination. We showed that (*i*) Gab1 is specifically expressed in OLs and oppositely regulated by triiodothyronine and PDGF; (*ii*) Gab1 is regulated by PDGF but not other growth factors in OLs; (*iii*) *Gab1* deletion in OLs causes hypomyelination in the CNS by reducing OPC differentiation; (*iv*) Gab1 binds to GSK3β and regulates its activity; and (*v*) Gab1 affects nuclear accumulation of β-catenin and regulates the expression of a number of factors critical to the transcription of myelin proteins. In summary, our work reveals a novel downstream target of PDGF signaling and an intrinsic cascade essential for OL development.

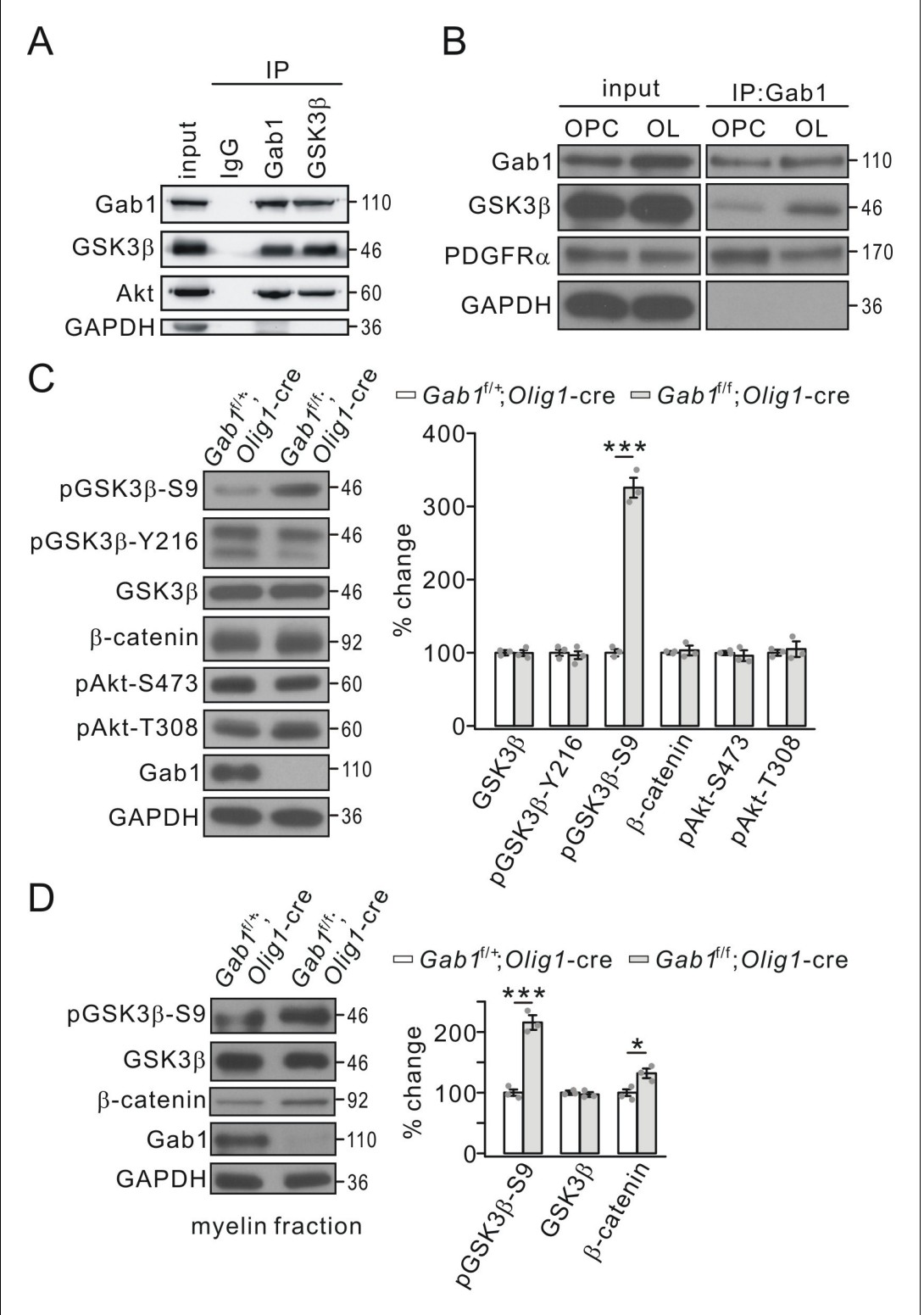

**Figure 7.** Gab1 binds to GSK3β and modulates its activity. (**A**) Precleared cortical lysates from wild-type mice (P21-23) were immunoprecipitated with mouse anti-Gab1 and anti-GSK3β antibodies. Immunoprecipitates were probed with antibodies to Gab1 (rabbit polyclonal antibody), GSK3β, Akt and GAPDH. The experiment was repeated three times. Rabbit IgG was used as the negative control. (**B**) The lysates of cultured OPCs and OLs were immunoprecipitated with mouse anti-Gab1 antibody. Immunoprecipitates were probed with antibodies to Gab1 (rabbit polyclonal antibody), GSK3β, PDGFRα, and GAPDH. The experiment was repeated three times. (**C**) The

*Figure 7 continued on next page*

*Figure 7 continued*

expression and phosphorylation of GSK3β, β-catenin and Akt in cerebral cortex from *Gab1*[f/+];*Olig1*-cre and *Gab1*[f/f];*Olig1*-cre mice at P21. GAPDH was the internal control. The increase of pGSK3β-S9 was 325 ± 16% (p=0.000077 *vs* control) in *Gab1*[f/f];*Olig1*-cre mice. *n* = 3/group. *t*-test, df = *t*(5). (D) Expression and phosphorylation of GSK3β and β-catenin in myelin fractions from *Gab1*[f/+];*Olig1*-cre and *Gab1*[f/f];*Olig1*-cre mice at P60. GAPDH was the internal control. Percentage changes of pGSK3β-S9 and β-catenin were 216 ± 12% (p=0.00036 *vs* control) and 133 ± 8% (p=0.013 *vs* control) in *Gab1*[f/f];*Olig1*-cre mice. *n* = 3/group. *t*-test, df = *t*(5). Gray dots indicate individual data points. *p<0.05. ***p<0.001.

## Expression of Gab proteins in the CNS

One important result in the present work is the expression locations of Gab1 and Gab2 in the CNS. Western blots indicated that Gab1 was absent from multiple brain regions of *Gab1*[f/f];*Olig1*-cre and *Gab1*[f/f];*Cspg2*-cre mice, suggesting its specific expression in OLs (*Figures 3* and *4*). Differently, Gab1 was found in both astrocytes and OLs in cultures (*Figure 1A*). These results appear contradictory but actually not, because the shaking procedure used in the purification of astrocytes and OLs does not completely separate two types of cells. Thus, cultured astrocytes might occur along with OLs and vice versa. Indeed, we were also able to detect Olig2, a marker protein of OLs, in astrocytic cultures (*Figure 1A*). The comparative western blots from *Gab1*[f/f];*Olig1*-cre and *Gab1*[f/f];*Campk2a*-cre mice also excluded the presence of Gab1 in neurons (*Figure 3B*), which was confirmed by the absence of Gab1 in cultured cortical neurons (*Figure 1A*).

The expression of Gab2 appeared different from that of Gab1, as it was found in neuronal, astrocytic and microglial cultures (*Figure 1A*). The differential expressions of two Gab proteins imply that they play distinct roles in the CNS, for example, Gab2 is not required for myelination (*Figure 4D*). It will be of interest to define the early expression of Gab1 and Gab2, two isoforms with similar structures, in neural progenitor cells. In fact, previous work has shown that Gab1 and Gab2 function differently in interacting with growth factors and modulating mitotic processes in neural progenitor cells (*Korhonen et al., 1999*; *Cai et al., 2002*; *Mao and Lee, 2005*).

## Relationships between Gab proteins and growth factor receptors in OLs

Gab1 and Gab2 are recognized as docking/scaffolding proteins of tyrosine kinase receptors (*Gu and Neel, 2003*). Growth factors receptors, one group of these receptors, play important roles in multiple cellular processes, including cell-cycle progression, differentiation, metabolism, survival, adhesion, motility, and migration, some of which are mediated by interacting with Gab proteins. For example, EGF, but not IGF and PDGF, increases the tyrosine phosphorylation of Gab1 and promotes the activity of Shp2 in epidermal cells (*Cai et al., 2002*; *Buonato et al., 2015*); Gab2 facilitates FGF-induced activation of Akt and decreases retinoic acid-induced apoptosis in embryonic stem cells (*Mao and Lee, 2005*). In OLs, we found that PDGF treatment induced a reduction in Gab1 expression, while other growth factors were ineffective (*Figure 2*). In addition, Akt phosphorylation was not changed by the conditional deletion of *Gab1* in OLs (*Figure 7*). These results imply that Gab1 is controlled by previously unknown machinery initiated by PDGF/PDGFRα signaling in OLs. PDGF activates PI3K and mitogen-activated protein kinase (MAPK)/extracellular regulated protein kinases (ERK) in OPC survival and migration (*Ebner et al., 2000*; *Vora et al., 2011*). While PDGF/PDGFRα signaling is important for migration, proliferation and myelination of OPCs, it remains unclear how it and its downstream targets, such as ERK, induce diverse effects. In this sense, our finding that PDGF reduces the expression of Gab1 provides new insight into the functions of PDGF signaling in OLs. We speculate that ERK might mediate PDGF-induced reduction in *Gab1* transcription in cultured OPCs, as ERK is known to affect many transcription factors.

## Modulation of GSK3β activity by Gab1

GSK3β plays key roles in neurogenesis, neuronal migration and axonal guidance (*Hur and Zhou, 2010*). We demonstrate that GSK3β is a positive regulator of OPC differentiation and is required for proper CNS myelination, but the upstream regulator of GSK3β was not addressed (*Zhou et al., 2014*). Signal-transduction studies have shown that GSK3β activity is regulated by two pathways. One is canonical Wnt pathway, where Axin2, a component of Wnt signaling, negatively modulates

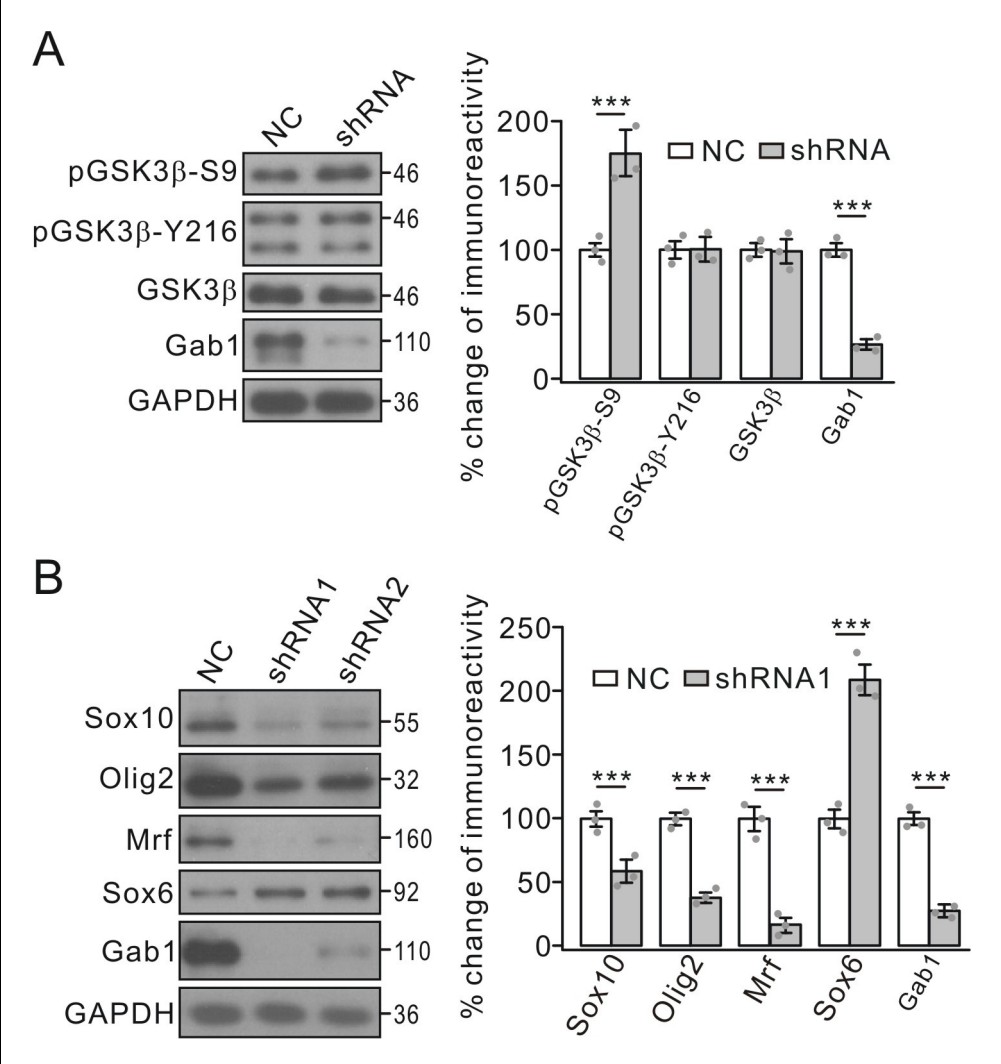

**Figure 8.** Effects of *Gab1*-knockdown on GSK3β and transcription factors. (**A**) OPC cultures were transfected with lentiviral vector-encoding with NC or shRNA, and the phosphorylation and expression of GSK3β were examined. GAPDH was the internal control. pGSK3β-S9p and Gab1 were 175 ± 16% (p=0.0063 *vs* control) and 27 ± 4% (p=0.00022 *vs* control) in shRNA compared to NC, respectively. n = 3/group. *t*-test, df = *t*(5). (**B**) Expressions of transcription factors in OPCs transfected with NC or shRNAs. Lysates were immunoblotted with antibodies against Sox10, Olig2, Mrf, Sox6 and Gab1. GAPDH was the control. Percentage changes in shRNA1 group are 58 ± 9% (Sox10) (p=0.0083 *vs* control), 38 ± 4% (Olig2) (p=0.0006 *vs* control), 16 ± 6% (Mrf) (p=0.00031 *vs* control), 209 ± 12% (Sox6) (p=0.00074 *vs* control), and 27 ± 3% (Gab1) (p=0.00022 *vs* control). n = 3/group. *t*-test, df = *t*(5). Gray dots indicate individual data points. ***p<0.001.

GSK3β activity (*Jho et al., 2002*). Another is PI3K/Akt pathway, which also inactivates GSK3β activity via inhibitory phosphorylation (*Cross et al., 1995*). Nevertheless, these pathways unlikely mediate the effects of the PDGF-Gab1 module in OL development, because there is no evidence for functional interaction between Wnt and PDGF signaling, and Akt activity was unchanged by conditional Gab deletion in OLs (*Figure 7*). Instead, we showed that Gab1 bound to GSK3β and *Gab1*-deficiency caused increased GSK3β phosphorylation at S9 (*Figure 7*). Based on these findings, we propose that Gab1 is a new regulator of GSK3β besides PI3K/Akt signaling. More importantly, our work shows that Gab1 links PDGFRα and GSK3β, two molecules critical for OL development but previously considered to be separated, providing new avenues to understand the complex intrinsic network during OL development.

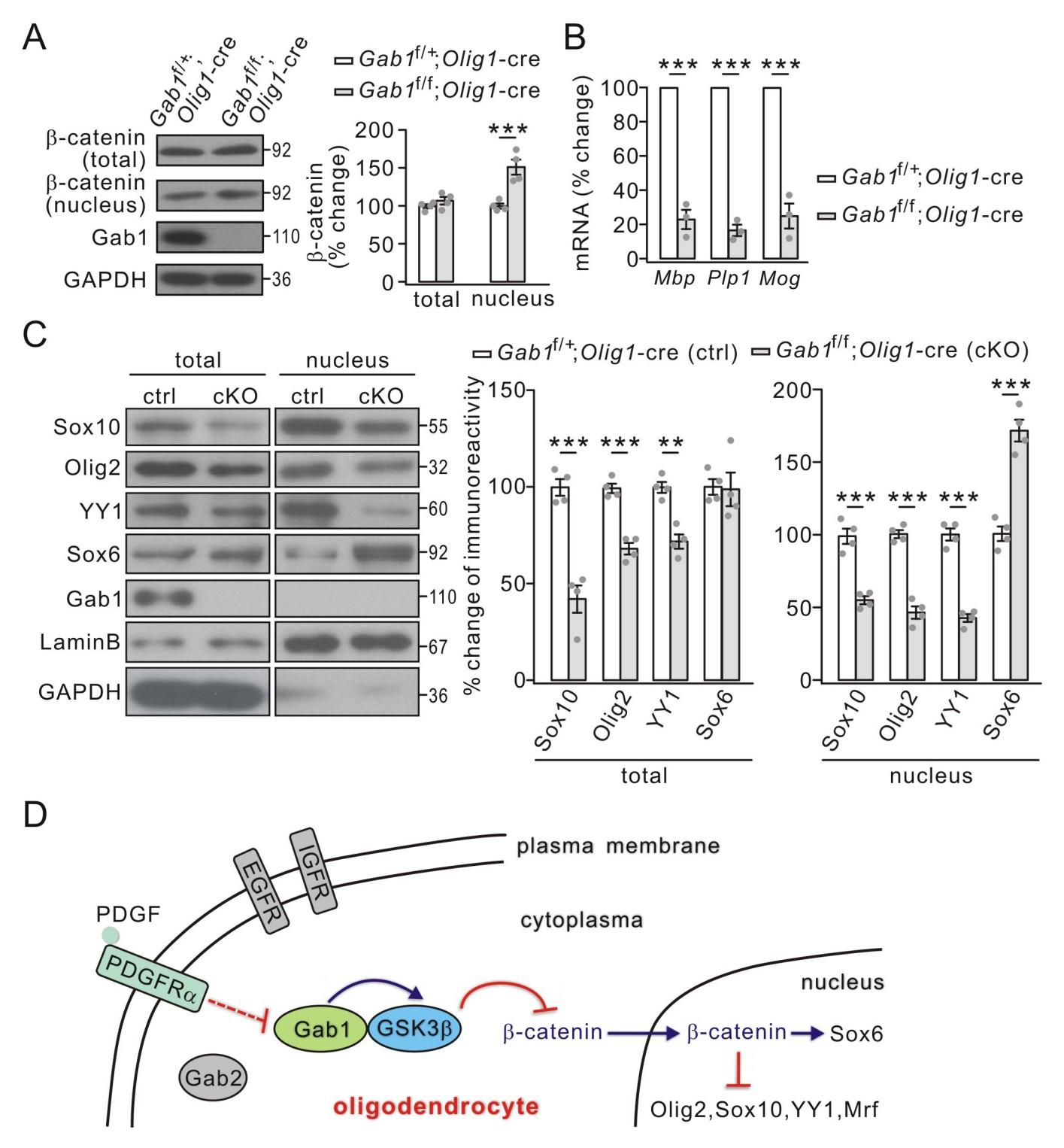

**Figure 9.** Effects of conditional *Gab1* deletion on β-catenin and transcription factors. (**A**) β-catenin expression in total and nuclear fractions from *Gab1*^f/+;*Olig1*-cre and *Gab1*^f/f;*Olig1*-cre mice at P21. GAPDH was the internal control. Percentage changes were 108 ± 6% (total; p=0.25) and 151 ± 11% (nuclear; p=0.0027) in *Gab1*^f/f;*Olig1*-cre mice. n = 4/group. *t*-test, df = *t*(7). (**B**) mRNA levels of myelin genes in *Gab1*^f/+;*Olig1*-cre and *Gab1*^f/f;*Olig1*-cre mice. *Gapdh* was the internal control. Percentage changes in *Gab1*^f/f;*Olig1*-cre group were 23 ± 7% (*Mbp*; p=0.00015 vs control), 34 ± 4% (*Plp1*; p=0.000013 vs control) and 25 ± 9% (*Mog*; p=0.00054 vs control). n = 3/group. *t*-test, df = *t*(5). (**C**) Total and nuclear transcription factors in *Gab1*^f/+; *Olig1*-cre (ctrl) and *Gab1*^f/f;*Olig1*-cre (cKO) mice at P21. Lysates were immunoblotted with antibodies against Sox10, Olig2, YY1, Sox6, Gab1, LaminB, and GAPDH. All proteins were normalized to GAPDH (total) or LaminB (nucleus) and percent changes are shown. For total, percentage changes were:
*Figure 9 continued on next page*

Figure 9 continued

42 ± 8% (Sox10; p=0.00044 vs control), 68 ± 4% (Olig2; p=0.0002 vs control), 72 ± 4% (YY1; p=0.0011 vs control), and 99 ± 10% (Sox6; p=0.9 vs control); p=vs control. For nucleus, percentage changes were 55 ± 3% (Sox10; p=0.00031 vs control), 47 ± 5% (Olig2; p=0.000041 vs control), 43 ± 3% (YY1; p=0.000025 vs control), and 172 ± 9% (Sox6; p=0.00023 vs control). n = 4/group. t-test, df = t(7). Gray dots indicate individual data points. (D) A working model illustrates the functions of Gab1 in OLs. Gab1 is specifically down-regulated by PDGF but not EGF or IGF. In WT condition, Gab1 binds to GSK3β and increases its activity, thereby inhibits nuclear accumulation of β-catenin and changes the expression of nuclear factors. As the result, Gab1 ablation leads to impaired OPC differentiation. Gray dots indicate individual data points. **p<0.01. ***p<0.001.

## Gab1 promotes OPC differentiation and CNS myelination

We found that the density of Olig2+PDGFRα+ OPCs did not differ between Gab1$^{f/f}$;Olig1-cre and Gab1$^{f/+}$;Olig1-cre mice (Figure 5). Moreover, Gab1-shRNA caused significant fewer MBP+ cells and lower MBP expression in cultured OLs than in naive control after triiodothyronine treatment (Figure 6). Therefore, we conclude that Gab1 ablation impairs OPC differentiation with little impact on the steady-state numbers of OPCs. Gab proteins provide a docking site for SH2 domain-containing signaling proteins, such as Shp2 and PI3K (Gu and Neel, 2003; Nishida and Hirano, 2003). Gab1 and Shp2 are required for normal differentiation in developing brain (Ahrendsen et al., 2018), but there are some differences between them. First, Shp2 mutants display phenotypes in controlling OPC proliferation (Zhu et al., 2010), whereas Gab1 mutation only affects OPC differentiation. Second, Shp2 mutants have thicker myelin sheaths (Ahrendsen et al., 2018) but Gab1 mutants did not (Figure 3D). Third, Shp2 loss leads to a delay in OPC differentiation (Ehrman et al., 2014; Ahrendsen et al., 2018), while Gab1 deficiency had more sustained effects. As both Gab1 and Shp2 transduce PDGFRα signal, these differences imply that OL development is subject to dynamic and multilevel regulations.

In keeping with our previous report (Zhou et al., 2014), we here show that Gab1 activates GSK3β and subsequently affects the expression of key transcription factors during OPC differentiation. Conditional knockout of Gab1 in OLs decreased the expressions of Sox10, Olig2, Mrf, and YY1, but increased the expression of Sox6 (Figures 7 and 8). It has been shown that Sox10, Olig2, Mrf and YY1 promote OPC differentiation (Emery, 2010), whereas Sox6 arrests it (Stolt et al., 2006). Thus, Gab1 functions in OPC differentiation and myelination by acting on both positive and negative transcription factors, similar to the functions of GSK3β in OPCs (Zhou et al., 2014).

Shin et al. (2014) reported that tyrosine phosphorylation of Gab1 in the sciatic nerves is up-regulated during the myelination period and conditional removal of Gab1 from Schwann cells (SCs) results in hypomyelination. The myelin defects in SC-specific conditional mutant (Gab1-SCKO) are not same as those in Gab1$^{f/f}$;Olig1-cre mice: Gab1-SCKO mice have fewer myelinated fibers, but the thickness of the myelin sheath is reduced and axonal diameter is unchanged (Shin et al., 2014). NRG-1, which is ineffective on Gab1 expression in cultured OLs (Figure 2), is responsible for inducing Gab1 effects in SCs (Shin et al., 2014). In consistent with our results, Shin et al. (2014) also found that Akt activity is not affected by Gab1 deletion in SCs. By comparison, we speculate that mitogens may act on axonal myelination through distinct mechanisms in the central and peripheral nervous systems.

It should be noted that Gab1 deletion in OLs causes partial hypomyelination: only approximately 50% decrease in the number of either myelinated axons (Figure 3B) or Olig2-positive OLs (Figure 5C) was found in Gab1$^{f/f}$;Olig1-cre mice. This phenotype suggests that, although Gab1 is important, other factors may play similar roles mediating PDGF/PDGFRα signaling during OPC development. Alternatively, not all OPCs require Gab1 for their differentiation. Indeed, Zheng et al. (2018) find that the development of a type of OPCs is independent of PDGFRα. If so, Gab1 may not regulate the development of these OPCs and thereby Gab1 deletion cannot eliminate myelin formation.

In conclusion, Gab1 is an important mediator of OPC differentiation and developmental myelination of the brain. We provide important insights into the mechanism of PDGF/PDGFRα signaling and propose that Gab1 acts as a promoting factor for CNS myelination process. Considering its unique expression, the roles of Gab1 in axonal remyelination and its therapeutic implications for demyelinating diseases such as multiple sclerosis, in which adult OPCs fail to differentiate effectively, might be expected.

## Materials and methods

All animal experiments were carried out in a strict compliance with protocols approved by the Animal Care and Use Committee at Zhejiang University School of Medicine.

### Animals

Mice were kept under temperature-controlled conditions on a 12:12 hr light/dark cycle with food and water ad libitum. $Gab1^{f/f}$, $Gab2^{-/-}$, and $Olig1$-cre mice were kindly provided by Gen-Sheng Feng at the Burnham Institute (Orlando, FL) and Q. Richard Lu at the Cincinnati Children's Hospital Medical Center (Cincinnati, OH). 2′,3′-cyclic nucleotide 3′-phosphodiesterase ($Cnp$)-cre, $Cspg2$ (NG2)-cre, calcium/calmodulin-dependent protein kinase II (CaMKII, $Campk2a$)-cre, $Nestin$-cre, proteolipid protein 1 ($Plp1$)-creER and $Pdgfra^{f/f}$ lines were purchased from the Jackson Laboratory (Bar Harbor, ME). The genetic background of all mice was C57BL/6. To produce conditional knockout mice ($Gab1^{f/f}$;$Olig1$-cre, $Gab1^{f/f}$;$Cspg2$-cre, $Gab1^{f/f}$;$Campk2a$-cre, $Gab1^{f/f}$;$Nestin$-cre, $Gab1^{f/f}$;$Plp1$-creER, and $Pdgfra^{f/f}$;$Cnp$-cre), heterozygous $Gab1^{f/+}$ mice obtained by crossing $Gab1^{f/f}$ mice with C57BL/6 mice and $Pdgfra^{f/f}$ mice were crossed with mice expressing a transgene encoding various cre recombinases. $Gab1^{f/+}$;$Olig1$-cre, $Gab1^{f/f}$, $Gab1^{f/+}$;$Plp1$-creER, $Pdgfra^{f/f}$, and C57BL/6 littermates served as controls in corresponding experiments. The resulting offspring were genotyped by PCR assays using their tail DNA. 4-Hydroxytamoxifen (tamoxifen) was dissolved in sunflower seed oil and administered to $Gab1^{f/+}$;$Plp1$-creER and $Gab1^{f/f}$;$Plp1$-creER at P180 by injections of 2 mg,>=8 hr apart for 5 days. All experiments were done in batches of mice of either sex.

### Antibodies and reagents

The antibodies against Olig2 (#ab9610; RRID:AB_570666), CC1 (#OP80; RRID:AB_2057371), MBP (#SMI99; RRID:AB_2140491), MOG (#MAB5680; RRID:AB_1587278), myelin associated glycoprotein (MAG; #MAB1567; RRID:AB_11214010), Mrf (#ABN45; RRID:AB_2750648), Sox10 (#AB5727; RRID: AB_11214438), NeuN (#MAB377; RRID:AB_2298772), Tuj1 (#MAB1637; RRID:AB_2210524), GFAP (#MAB360; RRID:AB_11212597), GAPDH (#MAB374; RRID:AB_2107445), and the black-gold staining kit (#AG105) were from Millipore. Antibodies against Gab1 (#3232; RRID:AB_2304999), Gab2 (#3239; RRID:AB_10698601), GSK3β (#9315; RRID:AB_490890), pGSK3β-S9 (#9323; RRID:AB_2115201), β-catenin (#8480; RRID:AB_11127855), pAkt-S473 (#4060; RRID:AB_2315049), and pAkt-T308 (#4056; RRID:AB_331163) were from Cell Signaling Technology. Antibodies against PDGFRα (#sc-380; RRID:AB_2263466), Yin-Yang 1 (YY1; #sc-1703; RRID:AB_2218501), and Lamin B (#sc-6216; RRID:AB_648156) were from Santa Cruz. The antibodies against pGSK3β-Y216 (#612312; RRID:AB_399627) and GSK3β (#610201,RRID:AB_397600) were from BD Bioscience. Antibodies against PLP (#ab28486; RRID:AB_776593) and Sox6 (#ab64946; RRID:AB_1143031) were from Abcam. PDGF-AA (#100-13A; RRID:AB_147954), EGF (#100–15; RRID:AB_147834), NRG-1 (#100–03), and insulin-like growth factor-1 (IGF-1; #100–11; RRID:AB_2737301) were from Peprotech. Horseradish peroxidase-conjugated secondary antibodies for immunoblotting were from Thermo Fisher Scientific (#31460, RRID:AB_228341; #31430, RRID:AB_228307). IgG antibody, Dulbecco's modified Eagle's medium (DMEM), Alexa Fluor-conjugated secondary antibodies, neurobasal, and B27 supplements were from Invitrogen. Proteinase inhibitor was from Merck Chemicals (Darmstadt, Germany). Other chemicals were from Sigma unless stated otherwise.

### Quantitative RT-PCR (qPCR)

mRNA was assessed by real-time PCR using an ABIPrism 7500 sequence detection system (Applied Biosystems). cDNA was synthesized by reverse transcription using oligo (dT) as the primer and proceeded to real-time PCR with gene-specific primers in the presence of SYBR Premix Ex Taq (TaKaRa, Dalian, China). Quantification was performed by the comparative cycle threshold (Ct) method, using $Gapdh$ as the internal control. The forward (F) and reverse (R) primers were: $Gab1$-F: 5′-CTG TCA GAG CAA GAA GCC-3′; $Gab1$-R: 5′-CAT ACA CCA TTT GCT GCT G-3′; $Gab2$-F: 5′-CTC TAC TTG CAC CAG TGC-3′; $Gab2$-R: 5′-CTC CAT TGA TAC AGT GTC C-3′; $Mbp$-F: 5′-GAG GCT TTT TGC AGA GGT T-3′; $Mbp$-R: 5′-CTC TGA GCT GCA GTT GGC-3′; $Plp1$-F: 5′-CTG GCT GAG GGC TTC TAC AC-3′; $Plp1$-R: 5′-GAC TGA CAG GTG GTC CAG GT-3′; $Mog$-F: 5′-AAA TGG CAA GGA CCA AGA TG-3′; $Mog$-R: 5′-AGC AGG TGT AGC CTC CTT CA-3′; $Gapdh$-F: 5′-GGT GAA GGT CGG TGT GAA CG-3′; and $Gapdh$-R: 5′-CTC GCT CCT GGA AGA TGG TG-3′.

## Astrocyte culture

Cortical astrocytes were cultured from embryonic SD rats (E18) according to *Ji et al. (2013)*. Cortices were dissected and incubated with trypsin-EDTA for 20 min at 37°C. Tissue was triturated and suspended in 10% DMEM. OLs and microglia were removed by shaking at 200 rpm for 2 hr at 37°C. Astrocytes were plated at a uniform density of $2 \times 10^5$ cells ml$^{-1}$.

## Neuronal culture

Cortical neurons from E16 SD rats were cultured according to previous work (*Wang et al., 2015*; *Zhou et al., 2018*). Dissociated neurons were plated and cultured in neurobasal supplemented with B-27 and L-alanyl-glutamine. Cultures were maintained at 37°C in a humidified incubator gassed with 95% $O_2$ and 5% $CO_2$.

## OPC culture

OPCs from SD rats (E18) were cultured according to previous work (*Zhou et al., 2014*; *Xie et al., 2018*). OPCs were collected from glial cultures by shaking for 1 hr at 200 rpm, incubating in fresh medium for 4 hr, and shaking at 250 rpm at 37°C for 16 hr. Collected OPCs were re-plated onto poly-D-lysine-coated plates and grown in neurobasal supplemented with 2% B27. PDGF-AA (10 ng/ml) was added to keep OPCs undifferentiated or triiodothyronine (40 ng/ml) was added for 3 days to allow differentiation.

## Lentiviral construction and transfection

Lentivirus encoding small hairpin RNA (shRNA) for *Gab1* (5'-GCG ATA GAT CCA GTT CCT TGG-3') was prepared by GenePharma (Shanghai, China). OPCs were transfected with GFP-tagged *Gab1*-shRNA or scrambled RNA, which were driven by U6 promoter, for 72 hr, and experiments were continued when > 60% of cultured OPCs were transfected judging by GFP fluroscence.

## Immunohistochemistry and immunocytochemistry

Sagittal sections (20 μm) were prepared and placed in a blocking solution (1% BSA, 0.3% Triton, 10% goat serum) for 1 hr at room temperature (RT). After washing with phosphate-buffered saline (PBS), sections were incubated sequentially with primary antibodies overnight at 4°C and secondary antibodies for 1 hr at RT. The secondary antibodies were diluted at 1:1000. The sections were mounted using ProLong Gold Antifade Reagent (Invitrogen). Cultured cells were fixed in 4% paraformaldehyde for 15 min at RT, washed with PBS and permeabilized in 0.2% Triton X-100 for 10 min, blocked in 10% BSA for 1 hr, and labeled with primary antibodies overnight at 4°C. Cells were then incubated with secondary antibodies (1:1000) for 1 hr at RT. All antibodies were diluted in PBS containing 1% BSA and 1% normal goat serum. The dilution ratios of primary antibodies were 1:1000 for MBP and 1:100 for Gab1, Olig2, CC1, and PDGFRα. For cell counts, four animals per genotype were used to examine cellular markers. Only the images of the midline of the corpus callosum were acquired in in vivo examination.

## Black-gold staining

Brain tissue was fixed in formalin and cut at 30 μm on a freezing sliding microtome. Tissue sections were hydrated and incubated in Black-gold solution at 60°C for 12 min. Staining was complete when the finest myelinated fibers turned to black. The sections were then rinsed in water, dehydrated in alcohols, and coverslipped with mounting medium.

## Transmission electron microscopy (TEM)

TEM was performed according to previous work (*Zou et al., 2011*; *Xie et al., 2018*). Ultra-thin sections were obtained using Ultracut UCT (Leica) and stained with uranyl acetate and lead citrate. Micrographs were captured in a Philips CM100 microscope (FEI).

## *g*-ratio analysis

TEM images containing large numbers of myelinated axons in cross-section were selected for *g*-ratio analysis. *g*-ratio analysis was performed with a threshold to identify axons and calculate their cross-sectional area (*Xie et al., 2018*), from which axon diameters were calculated using the formula for

the area of a circle, $A = \pi r^2$. An experimenter blinded to genotype then measured myelin sheath thickness of each axon, and excluded any improperly detected or obliquely cut axons from analysis.

## Coimmunoprecipitation (co-IP)

Cortices were lysed in RIPA buffer plus protease inhibitor. Protein concentrations were measured using BCA assays (Bio-Rad) after centrifugation at $16,000 \times g$ at 4°C for 10 min. Decimus supernatant was used for input and the remainder was used for IP. Precleared preparations were incubated with mouse anti-Gab1 antibody, which was precoupled to protein A-Sepharose beads (GE Healthcare) at 2–4 µg antibody/ml of beads for 2 hr in 50 mM Tris-HCl. Proteins on the beads were extracted with $2 \times$ SDS sample buffer and boiled for 5 min before western blotting.

## Myelin fraction isolation

According to previous work (*Saher et al., 2005*), crude myelin was obtained from brain homogenates by centrifuging at 25,000 rpm for 30 min and re-suspended in ice-cold water. The pellet was subjected to repeated centrifugations at 25,000 and 10,000 rpm, each for 15 min. The myelin pellets were then suspended in sucrose (0.32 and 0.85 M in order) and centrifuged at 25,000 rpm for 30 min. Myelin layers were suspended in 10 mM HEPES buffer (pH 7.4) with 1% Triton-X-100 for further experiments.

## Western blotting

Proteins derived from tissue or culture were rinsed with PBS and diluted in 1% SDS containing protease inhibitor cocktail. Protein concentration was determined using the BCA assay. Equal quantities of proteins were loaded onto sodium dodecyl sulfate-polyacrylamide gel (SDS-PAGE), transferred to PVDF membrane (Immobilon-P, Millipore), immunoblotted with antibodies, and visualized by enhanced chemiluminescence (Pierce Biotechnology). Primary antibody dilutions used were 1:200 for YY1; 1:500 for NeuN; 1:1000 for MAG, Mrf, Sox10, PDGFRα, Gab1, Gab2, GSK3β, pGSK3β-S9, pAkt-S473, and pAkt-T308; 1:2000 for Olig2, PLP and Lamin B; 1:5000 for MOG, Tuj1, and pGSK3β-Y216; 1:10,000 for MBP, CNP, GFAP, and β-catenin; and 1:20,000 for GAPDH. Film signals were digitally scanned and quantitated using ImageJ 1.42q (NIH).

## Statistics

The investigators who quantify western blots and immunostainings were blinded to the genotype. Data were analyzed using Excel 2003 (Microsoft), Igor Pro 6.0 (Wavemetrics), and SPSS 16.0 (SPSS). Sample sizes were constrained by availability of cohorts of age-matched mice and were not determined in advance. Statistical differences were determined using unpaired two-sided Student's *t*-test for two-group comparison or one-way ANOVA followed by Tukey's *post hoc* test for multiple group comparisons. For all analyses, the accepted level of significance was $p < 0.05$. '*n*' represents the number of animals or cultures tested. Data in the text and figures are presented as the mean $\pm$ SEM. The degree of freedom (df) was presented as df = $t(x)$ for *t* test or df = $F(v1, v2)$ for ANOVA.

## Acknowledgements

We thank Gen-Sheng Feng (Burnham Institute, La Jolla, CA) and Q Richard Lu (Cincinnati Children's Hospital Medical Center, Cincinnati, OH) for providing *Gab1*[f/f], *Gab2*[-/-], and *Olig1*-cre mice, the Core Facilities of Zhejiang University Institute of Neuroscience for technical assistance, and Iain C Bruce for reading the manuscript.

## Additional information

### Funding

| Funder | Grant reference number | Author |
| --- | --- | --- |
| Ministry of Science and Technology of the People's Republic of China | 2017YFA0104200 | Ying Shen |

| | | |
|---|---|---|
| National Natural Science Foundation of China | 31571051 | Liang Zhou |
| National Natural Science Foundation of China | 81625006 | Ying Shen |
| National Natural Science Foundation of China | 31820103005 | Ying Shen |
| Natural Science Foundation of Zhejiang Province | Z15C090001 | Ying Shen |
| Natural Science Foundation of Zhejiang Province | LQ17C090001 | Na Wang |
| Non-profit Central Research Institute Fund of Chinese Academy of Medical Sciences | 2017PT31038 | Ying Shen |
| Non-profit Central Research Institute Fund of Chinese Academy of Medical Sciences | 2018PT31041 | Ying Shen |
| Chinese Ministry of Education Project 111 Program | B13026 | Ying Shen |

The funders had no role in study design, data collection and interpretation, or the decision to submit the work for publication.

### Author contributions

Liang Zhou, Conceptualization, Data curation, Formal analysis, Funding acquisition, Validation, Investigation, Visualization, Methodology; Chong-Yu Shao, Data curation, Formal analysis, Validation, Investigation, Visualization, Methodology; Ya-Jun Xie, Si-Min Xu, Data curation, Formal analysis, Investigation, Methodology; Na Wang, Data curation, Formal analysis, Funding acquisition, Investigation, Methodology; Ben-Yan Luo, Zhi-Ying Wu, Resources, Validation, Visualization; Yue Hai Ke, Conceptualization, Resources, Validation, Visualization; Mengsheng Qiu, Conceptualization, Resources, Supervision, Project administration; Ying Shen, Conceptualization, Supervision, Funding acquisition, Validation, Visualization, Methodology, Writing - original draft, Project administration, Writing - review and editing

### Author ORCIDs

Na Wang https://orcid.org/0000-0002-1438-1508
Ying Shen https://orcid.org/0000-0001-7034-5328

### Ethics

Animal experimentation: All of the animals were handled according to approved protocol (ZJU20160019) of the Animal Experimentation Ethics Committee of Zhejiang University.

### Decision letter and Author response

Decision letter https://doi.org/10.7554/eLife.52056.sa1
Author response https://doi.org/10.7554/eLife.52056.sa2

## Additional files

### Supplementary files

• Transparent reporting form

### Data availability

All data generated or analysed during this study are included in the manuscript and supporting files.

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
