## [Decision Letter]

**Acceptance summary:**

The study identifies the scaffold protein, Gab1, as an intermediary between PDGF and oligodendrocyte function. Using conditional mice and a variety of assays for myelination and myelin specific proteins, the authors find a marked effect of Gab1 on oligodendrocyte differentiation and GSK3β signaling. Though Gab1 is used by multiple growth factors, Gab1 displays specificity during developmental myelination.

**Decision letter after peer review:**

Thank you for submitting your article "Gab1 mediates PDGF signaling and is critical to oligodendrocyte differentiation and CNS myelination" for consideration by *eLife*. Your article has been reviewed by 3 peer reviewers, and the evaluation has been overseen by a Reviewing Editor and Marianne Bronner as the Senior Editor. The following individuals involved in review of your submission have agreed to reveal their identity: Simon Murray (Reviewer #1); Charles D Stiles (Reviewer #2); Masakiyu Sasahara (Reviewer #3).

The reviewers have discussed the reviews with one another and the Reviewing Editor has drafted this decision to help you prepare a revised submission.

Summary:

This manuscript provides new insights into the regulatory mechanisms of oligodendrocytes. The reviewers noted a number of positive aspects of the study of Gab1 in myelination, considering its potential pivotal role in oligodendrocyte differentiation and downstream signaling. The effects of Gab1 are of interest because it may provide a missing link in oligodendrocyte biology. However, the reviewer comments indicated that several problems hindered the analysis with regards to the stage of myelination being analyzed and the strength of the analysis.

Essential revisions:

The major issues are (1) the issue of timing of myelination and requested later time points. This is a valid concern and that should be considered experimentally. (2) Another issue is the specificity of Gab1 as a key mediator of PDGF and the roles of GSK3 and S6 phosphorylation. (3) There were a number of experimental inconsistences in the measurements from Gab1 and PDGF conditional mice that need to be clarified. We are willing to consider a resubmission only if you can address these major concerns.

This manuscript provides novel insight into the myelinating process, identifying Gab1 as an important regulator of the myelinating process early in development. However, the vast majority of in vivo analyses are undertaken at p21 – at a time when myelination is developmentally very active (1-3). As such, the phenotype observed could simply reflect a slight developmental delay to the otherwise normal process of myelination, rather than being 'critical to oligodendroglial development and myelination.' Unless later time points are assessed that demonstrate the persistence of the cellular and myelin phenotype, I cannot support the conclusion that Gab1 is critical to these processes.

1) There are substantial reservations about the robust nature of the in vivo data as presented. The conclusions drawn about the critical role Gab1 plays in myelination in vivo are largely drawn from Figures 3B-E. I have concerns about the TEM images and quantitation in Figure 3B, the robustness of the g-ratio analysis in Figure 3C and Figure 3D and E are qualitative at best. Given that these in vivo analyses are all from animals aged p21, it is essential to look at later developmental time points to see if the hypomyelinating phenotype persists to support the assertion of a 'critical' role in myelination.

2) The data supporting the assertion that Gab1 is critical for oligodendroglial development in vivo is largely drawn from Figure 5A-D. I have several concerns about how these cells were identified and the method of their quantitation. The data is presented as number of cells /section – this is highly unusual, normally a density (cells/mm^2^) is presented. This is important, as it allows the reader to benchmark this quantitation against other publications that have undertaken the same analyses, albeit in different mice, again to assess how robust the data is. Again, these in vivo analyses are all from animals aged p21. It is essential to look at later developmental time points to see if the reduced OL phenotype persists to support the assertion of a 'critical' role in oligodendrocyte development.

3) Ultimately both the in vivo myelin and cellular phenotypes are partial – approximately 50% decrease in OLs and 50% decrease in myelin, yet the deletion of Gab1 is assessed as being complete. So why is the phenotype only partial? If Gab1 was critical for oligodendrocyte differentiation and CNS myelination, then surely the phenotype would have been more penetrant. On face value the data indicate that Gab1 is only critical for the differentiation and myelination of some oligodendrocytes, but this is not discussed. Either is the potential that Gab1 is important for OL survival, not only differentiation.

4) The data showing the mechanism of Gab1 action is intriguing. However, the data are confusing. The assertion is that GSK3β phosphorylation at S9 correlates with the differentiation of OPCs and myelin gene expression (subsection “Gab1 binds to GSK3β and modulates its activity” and in the model presented in Figure 9D). Yet the data are showing increased S9 phosphorylation in both cKO mice (Figure 7) and following knock-down in vitro (Figure 8) which leads to reduced OL differentiation and myelination. This appears counter-intuitive.

The central claims of this article from Zhou et al., are (1) Gab1, but not Gab2 is expressed in oligodendrocytes, (2) Gab1 expression is regulated by PDGF but not by other growth factors, (3) Gab1 deletion causes hypo myelination by opposing the differentiation of oligodendrocyte progenitors, (4) Gab1 binds to GSK3β and regulates its activity and (5) Gab1 regulates nuclear accumulation of b-catenin and thereby regulates transcription of nuclear factors that are critical to transcription of myelin proteins. Claims 1 through 3 are well supported by the data and would be of interest to a broad community of investigators in the areas of developmental neurobiology and demyelination disease states. Data supporting claims 4 and 5 are less solid in my view. Concerns here are as follows:

Claim #4.

(1) Figures 7A, B show that anti-Gab1 antibody will pull down GSK3, AKT and PDGFRα from cortical tissues, as well as oligodendrocytes and oligodendrocyte progenitors. Are these co-IPs symmetrical, e.g. will anti-GSK3 pull down Gab1? A reverse IP would help to preclude adventitious interactions and solidify the argument for a Gab1:GSK3 complex.

(2) I find then figures and text dealing with regulation of GSK3 by Gab1 very confusing. Part of the problem is with the text dealing with GSK3 activation. In subsection “Gab1 binds to GSK3β and modulates its activity” the authors tell us "…phosphorylations at S9 and Y216 are positive regulators of GSK3β activity". Is phosphorylation at both positions required to activate GSK3? I assume that this is not the case because the authors go on to show us that only S9 is phosphorylated in Gab1-null cells. However, nowhere does it say that S9 phosphorylation alone is sufficient to activate GSK3β. The authors should correct this ambiguity because it is critical to their model…and speaking of that model, the schematic figure (Figure 9D) is confusing and (to my mind) misleading. In particular, the data showing that knockout or knockdown of Gab1 leads to an increase of phosphorylated GSK3β would suggest the following model:

PDGFRα--| Gab1--| GSK3β(Σ9) →intranuclear β-catenin

This seems to be the model that the authors intend. However, the model as drawn in Figure 9D would indicate the following:

PDGFRα--| Gab1→GSK3β(Σ9) →intranuclear β-catenin

Claim #5. The effects of Gab1 knockout on expression of transcription factors that regulate myelination are persuasive. However, the immunoblot data supporting the notion that knockout of Gab1 promotes an increase in intranuclear β-catenin are fairly nuanced – only a 50% increase as shown in Figure 9A. Would an immunostain for β-catenin be more persuasive?

---

## [Author Response]

Essential revisions:1) the issue of timing of myelination and requested later time points. This is a valid concern and that should be considered experimentally.

We agree. In the new experiments, we examined the myelination at a later developmental stage. Our results obtained from TEM, Western blots, and IHC support that Gab1 deletion also impairs myelination in the 3-month-old cKO mice. Detailed explanation is given in the reply to reviewer comments #1.

2) Another issue is the specificity of Gab1 as a key mediator of PDGF and the roles of GSK3 and S6 phosphorylation.

To the first question, our answer is that Gab1 is an important but not the sole factor of PDGF signaling, which is also given in the response to reviewer comments #3. Accordingly, we add a paragraph in subsection “Gab1 promotes OPC differentiation and CNS myelination” to discuss the partial impairment in *Gab1*^f/f^;*Olig1*-cre mice: (1) Other effectors may also mediate PDGF/PDGFRα signaling during OPC differentiation; (2) The development of a type of OPCs is independent of PDGF/PDGFRα signaling. Moreover, we change the word “critical” to “essential” in the manuscript title.

To the second question, the confusion is ascribed to our wrong writing about GSK3β-S9 in the original subsection “Gab1 binds to GSK3β and modulates its activity”. The correct description should be “the phosphorylation at S9 is a negative regulator while the phosphorylation at Y216 is a positive regulator of GSK3β activity (Wang et al., 1994; Hughes et al., 1993)”, however we wrote “…S9 is a positive regulator…” by mistake. It is known that the S9-phosphorylation of GSK3β regulates the nuclear accumulation of β-catenin. This mistake was due to our carelessness in writing and all authors sincerely apologize for this mistake. This answer is also given in the response to reviewer comments #4 and #6.

3) There were a number of experimental inconsistences in the measurements from Gab1 and PDGF conditional mice that need to be clarified.

We sincerely apologize for all mistakes and inconsistences, including “S9 positive regulator”, “PDGF^f/f^”, misplaced statistical numbers, wrong labeling in figures, and so on. Our responses to these mistakes are given in the responses to reviewer comments.

1) There are substantial reservations about the robust nature of the in vivo data as presented. The conclusions drawn about the critical role Gab1 plays in myelination in vivo are largely drawn from Figures 3B-E. I have concerns about the TEM images and quantitation in Figure 3B, the robustness of the g-ratio analysis in Figure 3C and Figure 3D and E are qualitative at best. Given that these in vivo analyses are all from animals aged p21, it is essential to look at later developmental time points to see if the hypomyelinating phenotype persists to support the assertion of a 'critical' role in myelination.

Thank you for these insightful comments.

In order to investigate if the hypomyelinating phenotype persists at later developmental stages, we examined the myelin structure in 3-month-old mice. The TEM pictures and *g*-ratio analysis are presented in Figures 3F-G. We also performed MBP staining in the cortex, corpus callosum and the hippocampus from 3-month-old control mice (Figure 3H). Moreover, we examined the expression of myelin-related proteins (MBP, CNP and MOG) in cKO mice at this age and the data are presented in Figure 4—figure supplement 1. All data support that Gab1 deletion impairs myelination at the late stage of adult mice.

We are very sorry for the wrong information of TEM images. The pictures were actually derived from optic nerve. We corrected the legend for Figure 3B “TEM images from optic nerve of …” and also mark it inside Figure 3B.

To increase the robustness of analysis, we recruited 160 axons from either control (*n* = 4) and cKO (*n* = 4) group into the analysis of the percentage of myelinated axons and *g*-ratio (see figure legend of Figures 3B-C). Our results indicated fewer myelinated axons and unchanged *g*-ratio of myelin sheathes in *Gab1*^f/f^;*Olig1*-cre mice (Figures 3B-C).

2) The data supporting the assertion that Gab1 is critical for oligodendroglial development in vivo is largely drawn from Figure 5A-D. I have several concerns about how these cells were identified and the method of their quantitation. The data is presented as number of cells /section – this is highly unusual, normally a density (cells/mm2) is presented. This is important, as it allows the reader to benchmark this quantitation against other publications that have undertaken the same analyses, albeit in different mice, again to assess how robust the data is. Again, these in vivo analyses are all from animals aged p21. It is essential to look at later developmental time points to see if the reduced OL phenotype persists to support the assertion of a 'critical' role in oligodendrocyte development.

We thank these excellent comments.

As the response, first we performed Olig2/CC1 and Olig2/PDGFRα double staining in the cerebral cortex and the corpus callosum of *Gab1*^f/f^;*Olig1*-cre and *Gab1*^f/+^;*Olig1*-cre mice at P21. Cells positive to Olig2 and CC1 are recognized as differentiated OLs and cells positive to PDGFRα and Olig2 are recognized as OPCs (see the text in subsection “*Gab1* ablation reduces OPC differentiation” and figure legends for Figures 5A-B).

Second, we used the density (cells/mm^2^) to present the results of cell counting and the data are shown in Figure 5C and its legend.

Third, we examined the reduced OL phenotype in the cerebral cortex and the corpus callosum of *Gab1*^f/f^;*Olig1*-cre and *Gab*1^f/+^;*Olig1*-cre mice at the age of 3 month. We found that differentiated OLs decreased in *Gab1*^f/f^;*Olig1*-cre mice at this age (see Figure 5—figure supplement 1).

3) Ultimately both the in vivo myelin and cellular phenotypes are partial – approximately 50% decrease in OLs and 50% decrease in myelin, yet the deletion of Gab1 is assessed as being complete. So why is the phenotype only partial? If Gab1 was critical for oligodendrocyte differentiation and CNS myelination, then surely the phenotype would have been more penetrant. On face value the data indicate that Gab1 is only critical for the differentiation and myelination of some oligodendrocytes, but this is not discussed. Either is the potential that Gab1 is important for OL survival, not only differentiation.

We thank this insightful comment. Therefore, we add a paragraph to subsection “Gab1 promotes OPC differentiation and CNS myelination” to discuss the possibilities for the partial phenotype, which is “It should be noted that Gab1 deletion in OLs causes partial hypomyelination: only approximately 50% decrease in the number of either myelinated axons (Figure 3B) or Olig2-positive OLs (Figure 5C) was found in Gab1^f/f^;Olig1-cre mice. This phenotype suggests that, although Gab1 is important, other factors may play similar roles mediating PDGF/PDGFRα signaling during OPC development. Alternatively, not all OPCs require Gab1 for their differentiation. Indeed, Zheng et al., (2018) find that the development of a type of OPCs is independent of PDGFRα. If so, Gab1 may not regulate the development of these OPCs and thereby Gab1 deletion cannot eliminate myelin formation”. In addition, we change the word “critical” in the manuscript title and text to “essential” or “important”.

4) The data showing the mechanism of Gab1 action is intriguing. However, the data are confusing. The assertion is that GSK3β phosphorylation at S9 correlates with the differentiation of OPCs and myelin gene expression (subsection “Gab1 binds to GSK3β and modulates its activity” and in the model presented in Figure 9D). Yet the data are showing increased S9 phosphorylation in both cKO mice (Figure 7) and following knock-down in vitro (Figure 8) which leads to reduced OL differentiation and myelination. This appears counter-intuitive.

We apologize for a mistake regarding this comment: The description about GSK3β-S9 in the original subsection “Gab1 binds to GSK3β and modulates its activity” was wrong. We wrote “…S9 is a positive regulator…” by mistake. Actually, the correct sentence should be “the phosphorylation at S9 is a negative regulator while the phosphorylation at Y216 is a positive regulator of GSK3β activity (Wang et al., 1994; Hughes et al., 1993)”. This mistake was due to bad communications during manuscript polishing. The S9-phosphorylation GSK3β is important because it regulates the nuclear accumulation of β-catenin.

The central claims of this article from Zhou et al., are (1) Gab1, but not Gab2 is expressed in oligodendrocytes, (2) Gab1 expression is regulated by PDGF but not by other growth factors, (3) Gab1 deletion causes hypo myelination by opposing the differentiation of oligodendrocyte progenitors, (4) Gab1 binds to GSK3β and regulates its activity and (5) Gab1 regulates nuclear accumulation of b-catenin and thereby regulates transcription of nuclear factors that are critical to transcription of myelin proteins. Claims 1 through 3 are well supported by the data and would be of interest to a broad community of investigators in the areas of developmental neurobiology and demyelination disease states. Data supporting claims 4 and 5 are less solid in my view. Concerns here are as follows:Claim #4.(1) Figures 7A, B show that anti-Gab1 antibody will pull down GSK3, AKT and PDGFRα from cortical tissues, as well as oligodendrocytes and oligodendrocyte progenitors. Are these co-IPs symmetrical, e.g. will anti-GSK3 pull down Gab1? A reverse IP would help to preclude adventitious interactions and solidify the argument for a Gab1:GSK3 complex.

We agree to this comment. We re-performed Co-IP experiment using antibodies to Gab1 and GSK3β. As shown in Figure 7A, GSK3β can also pull down Gab1, further confirming the interaction between Gab1 and GSK3β.

(2) I find then figures and text dealing with regulation of GSK3 by Gab1 very confusing. Part of the problem is with the text dealing with GSK3 activation. In subsection “Gab1 binds to GSK3β and modulates its activity” the authors tell us "…phosphorylations at S9 and Y216 are positive regulators of GSK3β activity". Is phosphorylation at both positions required to activate GSK3? I assume that this is not the case because the authors go on to show us that only S9 is phosphorylated in Gab1-null cells. However, nowhere does it say that S9 phosphorylation alone is sufficient to activate GSK3β. The authors should correct this ambiguity because it is critical to their model…and speaking of that model, the schematic figure (Figure 9D) is confusing and (to my mind) misleading. In particular, the data showing that knockout or knockdown of Gab1 leads to an increase of phosphorylated GSK3β would suggest the following model:PDGFRα--| Gab1--| GSK3β(Σ9) →intranuclear β-cateninThis seems to be the model that the authors intend. However, the model as drawn in Figure 9D would indicate the following:PDGFRα--| Gab1→GSK3β(Σ9) →intranuclear β-catenin

Again, we apologize for the mistake: the description about GSK3β-S9 in the original subsection “Gab1 binds to GSK3β and modulates its activity” was wrong. The correct sentence should be “the phosphorylation at S9 is a negative regulator while the phosphorylation at Y216 is a positive regulator of GSK3β activity (Wang et al., 1994; Hughes et al., 1993)”, which is now added in subsection “Gab1 binds to GSK3β and modulates its activity”. Hence, the increased S9 phosphorylaiton of GSK3β (Figure 7C) indicates that GSK3β activity is inhibited in cKO mice.

We are also sorry for the ambiguity in the model figure. We actually want to suggest “PDGFRα --| Gab1 → GSK3β --| intranuclear β-catenin”. To make our meaning clearer, we have modified the model figure and its legend according to WT condition.

Claim #5. The effects of Gab1 knockout on expression of transcription factors that regulate myelination are persuasive. However, the immunoblot data supporting the notion that knockout of Gab1 promotes an increase in intranuclear β-catenin are fairly nuanced – only a 50% increase as shown in Figure 9A. Would an immunostain for β-catenin be more persuasive?

We thank this comment. We ever performed β-catenin staining in the brain. As the result, we found the staining showed a diffuse expression in the whole brain and there was no obvious β-catenin signal in the nucleus.

We examined the expression of β-catenin in isolated sub-cellular fractions of WT animals using blots. As shown in Author response image 1, we found that β-catenin is significantly higher in the cytoplasm than in the nucleus, and the nuclear percentage from the total was very little. This result from the other side confirms the difficult in imaging β-catenin in the nucleus.

We hope these answers meet the reviewer’s satisfaction.

**Author response image 1. respfig1:** The expression of β-catenin in the cytoplasma and nucleus.